# Executioner caspase is proximal to Fasciclin 3 which facilitates non-lethal activation in *Drosophila* olfactory receptor neurons

Masaya Muramoto[1†], Nozomi Hanawa[1†], Misako Okumura[2,3], Takahiro Chihara[2,3], Masayuki Miura[1,4]*, Natsuki Shinoda[1]*

[1]Department of Genetics, Graduate School of Pharmaceutical Sciences, The University of Tokyo, Tokyo, Japan; [2]Program of Biomedical Science, Graduate School of Integrated Sciences for Life, Hiroshima University, Higashi-Hiroshima, Hiroshima, Japan; [3]Program of Basic Biology, Graduate School of Integrated Sciences for Life, Hiroshima University, Higashi-Hiroshima, Hiroshima, Japan; [4]Laboratory for Cell Vigor Regulation, National Institute for Basic Biology, Okazaki, Japan

*For correspondence:
miura@nibb.ac.jp (MM);
f-shinoda.natsuki@g.ecc.u-tokyo.ac.jp (NS)

[†]These authors contributed equally to this work

Competing interest: The authors declare that no competing interests exist.

## eLife Assessment

This **important** study identifies a mechanism by which caspases are activated in a non-lethal context to induce functional modulation in *Drosophila* olfactory receptor neurons. To deliver, the authors generated a new reporter of caspases, used TurboID to identify proteins proximal of the Drosophila executioner caspases Drice, and then focused on Fasciclin 3 as a mediator. The experimental results and the main conclusions are **convincing**. This substantial body of work will be of interest to researchers across fields, from neuroscience of olfaction to development and cell biology.

**Abstract** The nervous system undergoes functional modification independent of cell turnover. Caspase participates in reversible neuronal modulation via non-lethal activation. However, the mechanism that enables non-lethal activation remains unclear. Here, we analyzed proximal proteins of *Drosophila* executioner caspase in the adult brain using TurboID. We discovered that executioner caspase Drice is, as an inactive proform, proximal to cell membrane proteins, including a specific splicing isoform of cell adhesion molecule Fasciclin 3 (Fas3), Fas3G. To investigate whether sequestration of executioner caspase to plasma membrane of axons is the mechanism for non-lethal activation, we developed a Gal4-Manipulated Area-Specific CaspaseTracker/CasExpress system for sensitive monitoring of caspase activity near the plasma membrane. We demonstrated that *Fas3G* overexpression promotes caspase activation in olfactory receptor neurons without killing them, by inducing expression of initiator caspase Dronc, which also comes close to Fas3G. Physiologically, *Fas3G* overexpression-facilitated non-lethal caspase activation suppresses innate olfactory attraction behavior. Our findings suggest that subcellularly restricted caspase activation, defined by caspase-proximal proteins, is the mechanism for non-lethal activation, opening the methodological development of reversible modification of neuronal function via regulating caspase-proximal proteins.

**eLife digest** Networks of neurons in the nervous system are continuously adapting and changing their role, often without having to kill and replace cells. A group of enzymes called caspases – which are best known for initiating cell death – are thought to play a significant role in this process.

Studies have shown that caspases contribute to shaping neural connections and are involved in brain functions such as learning and memory. However, it remains unclear how the enzymes safely perform these tasks without mistakenly killing neurons.

To investigate this question, Muramoto, Hanawa et al. employed a technique called TurboID to label caspases and their nearby proteins in the brains of fruit flies. Their analysis showed that caspases are typically located close to proteins in the membrane that surrounds the interior of a cell.

Muramoto, Hanawa et al. then increased the production of one of these membrane proteins, called Fas3G, in olfactory neurons which are responsible for detecting smells. This increased the activity of caspases without triggering cell death. It also reduced how attracted the flies were to odors they are normally drawn to, suggesting that non-lethal caspase activation can suppress the activity of olfactory neurons.

These results suggest that proteins that confine caspases to specific regions of a cell, like the membrane, can activate the enzyme without triggering cell death. This localized, non-lethal activation lets neurons temporarily adjust their function, allowing the nervous system to adapt without needing to destroy and replace cells. The findings also highlight how important location is in guiding the activity of caspases, and it is possible that this concept may apply to other enzymes.

## Introduction

During and even after development, the nervous system undergoes functional modification usually not depending on cell turnover. Caspase, best known as a cysteine aspartic acid protease involved in cell death, participates in neuronal functional modulation through non-lethal activation (*Mukherjee and Williams, 2017*). In zebrafish embryos, non-lethal caspase activity promotes axon arbor dynamics in retinal ganglion cells (*Campbell and Okamoto, 2013*). In *Drosophila*, caspase activation promotes engulfment of pruned dendrites during metamorphosis (*Williams et al., 2006*). Furthermore, non-lethal caspase activity promotes the maturation of olfactory sensory neurons during development in mice (*Ohsawa et al., 2010*). In the mature nervous system, non-lethal caspase activity induces long-term depression by AKT cleavage in mice (*Li et al., 2010*). These processes are termed caspase-dependent non-lethal cellular processes (CDPs) (*Aram et al., 2017*). While a growing number of CDPs have been identified (*Nakajima and Kuranaga, 2017*; *Shinoda and Miura, 2024*), relatively less is known about its regulatory mechanisms that enable specific non-lethal functions without causing cell death, especially in nervous systems (*Mukherjee and Williams, 2017*). One potential mechanism is the restriction of the extent and spread of activated caspases (*Mukherjee and Williams, 2017*), yet experimental evidence is limited.

Proteins that interact with initiator caspases are among the key regulators of CDPs. Tango7, the *Drosophila* translation initiation factor eIF3 subunit m, acts as an adapter and regulates the localization of the initiator caspase Dronc. During sperm individualization, Tango7 directs Dronc to the individualization complex (*D'Brot et al., 2013*). In salivary glands, Tango7 directs Dronc to the plasma membrane (*Kang et al., 2017*). In both cases, caspase activity promotes cellular remodeling without inducing cell death (*D'Brot et al., 2013*; *Kang et al., 2017*). CRINKLED, a *Drosophila* unconventional myosin, functions as an adaptor of Dronc and facilitates the cleavage of Shaggy46 (*Orme et al., 2016*). Shaggy46, activated by caspase-mediated cleavage, suppresses the emergence of ectopic sensory organ precursors (*Kanuka et al., 2005*). During apoptosis-induced proliferation (AiP), Myo1D, another *Drosophila* unconventional myosin, facilitates Dronc translocation to the basal side of the plasma membrane of imaginal discs to promote AiP (*Amcheslavsky et al., 2018*). Thus, initiator caspase-interacting proteins regulate caspase activity in CDPs. However, given the importance of executioner caspases, which are activated by the initiator caspase to cleave a broad range of substrates and are essential for apoptosis (*Julien and Wells, 2017*; *Kumar, 2007*), little is known about the subcellular compartmentalization and interacting proteins of those

involved in CDPs. This lack of understanding is partly due to the absence of protein-protein interaction domains (e.g. DED and CARD) in the N-terminus of executioner caspases (*Kumar et al., 2022*).

To understand the molecular mechanism regulating caspase activity involved in CDPs, we focused on the proximal proteins of executioner caspases using TurboID, a proximity labeling technique that analyzes protein proximity in vivo (*Branon et al., 2018*; *Qin et al., 2021*). We have previously established C-terminal TurboID knocked-in caspase fly lines for Dronc (initiator), Drice (executioner), and Dcp-1 (executioner) (*Shinoda et al., 2019*). Importantly, we have demonstrated that one of the preferred substrates of Drice, BubR1, is proximal to Drice in vivo (*Shinoda et al., 2023*), suggesting that substrate preference is exploited by protein proximity. Therefore, the proximity of executioner caspases to proteins is emerging as a means of regulating specific non-lethal functions.

To investigate the regulatory mechanisms of non-lethal function of caspase in the nervous system, we conducted a comprehensive analysis of proximal proteins of executioner caspase Drice. We discovered that Drice is, as an inactive proform, primarily proximal to cell membrane proteins, including cell adhesion molecule Fasciclin 3 (Fas3). Notably, Drice is proximal to the specific alternative splicing isoforms of Fas3, Fas3G. To ascertain whether Fas3G modulates caspase activity, we developed a Gal4-Manipulated Area Specific CaspaseTracker/CasExpress (MASCaT) system, which permits the monitoring of caspase activity near plasma membrane with high sensitivity and simultaneous genetic manipulation in the cells of interest. Using MASCaT, we demonstrated that *Fas3G* overexpression enhances caspase activation without killing olfactory receptor neurons (ORNs) through the induction of the expression of initiator caspase Dronc, which also comes proximal to Fas3G. Subsequently, we showed that *Fas3G* overexpression-facilitated non-lethal caspase activation in ORNs suppresses innate olfactory attraction behavior. Collectively, our findings suggest that caspase activation is subcellularly restricted, the platform of which is defined by caspase-proximal proteins, for non-lethal functions. In contrast to lethal activation, suppressing neuronal activity with non-lethal caspase activation is reversible. By regulating caspase-proximal proteins, reversible modification of neuronal function by non-lethal caspase activation will be achieved.

## Results

### Drice is the major executioner caspase expressed in the adult brains

Apoptotic stimuli activate initiator caspases that trigger a proteolytic cascade, leading to the activation of executioner caspases and culminating in apoptosis (*Green, 2019*). In *Drosophila*, the caspase-mediated apoptotic signaling pathway is highly conserved (*Figure 1A*; *Nakajima and Kuranaga, 2017*). The activity of executioner caspases, specifically Drice and Dcp-1, is regulated by apoptotic signaling pathways, including the initiator caspase Dronc (*Figure 1A*). To investigate the expression patterns of *Drosophila* caspases, we employed *Caspase::V5::TurboID* knocked-in fly lines (*Shinoda et al., 2019*). We found low levels of expression of the initiator caspase Dronc in adult heads (*Figure 1B*). Conversely, we observed high expression of the executioner caspase Drice, whereas Dcp-1 was expressed at lower levels in adult heads (*Figure 1B*). The proximal proteins of each caspase were labeled with TurboID in adult heads by the administration of 100 μM biotin (*Figure 1B*). Furthermore, we investigated the expression patterns of caspases using histochemical analysis by examining streptavidin (SA) signal. Consistent with the western blotting findings, Drice showed higher expression levels than Dronc and Dcp-1, which were expressed at lower levels (*Figure 1C*). We found that Drice was expressed in distinct regions, including the mushroom bodies (MBs), subesophageal ganglions (SOGs), optic lobes (OLs), and antennal lobes (ALs), which showed the highest levels of expression (*Figure 1C*). The *Drosophila* ALs comprise synaptic contacts between the axons of ORNs, the dendrites of projection neurons, and local interneuron processes. Neuron-type-specific Gal4s were used to knock down *Drice,* and the expression patterns were subsequently examined. We found that Drice was expressed predominantly in ORNs in the AL (*Figure 1D*). These results show that Drice is highly expressed in a specific subset of neurons in the adult brains. Comparisons of the expression patterns of Drice were consistent across both males and females (*Figure 1—figure supplement 1A and B*), indicating an absence of sex-based differences, which justifies the use of males for functional simplicity in this study.

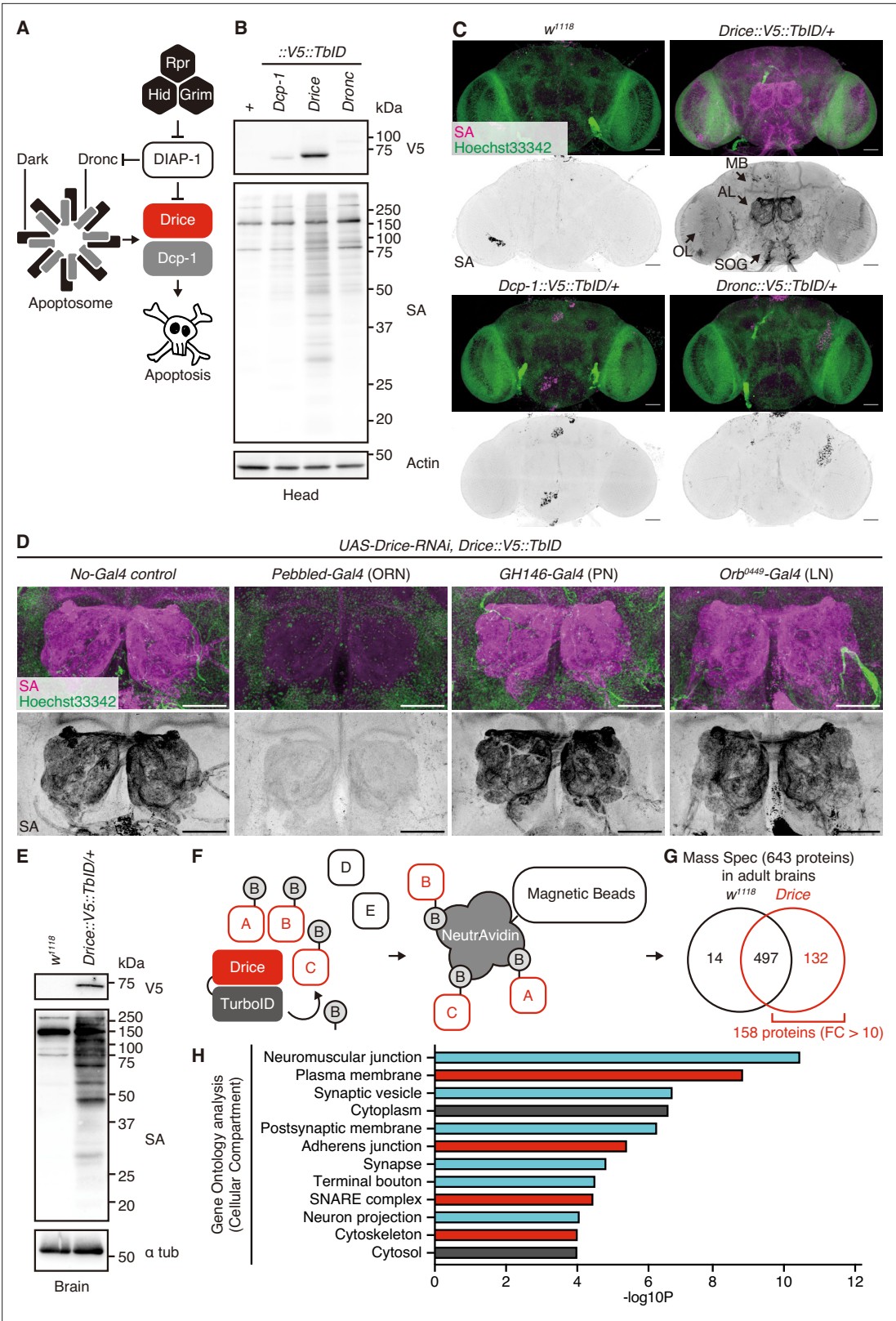

**Figure 1.** Expression patterns and proximal proteins of Drice in the adult brain. (**A**) A schematic diagram of *Drosophila* apoptosis signaling. (**B**) Western blot of expression of C-terminally V5::TurboID knocked-in tagged caspases in adult male heads. Biotinylated proteins are detected by streptavidin (SA). (**C**) Representative images of adult male brains. Expression patterns of each C-terminally V5::TurboID knocked-in tagged caspase are visualized using SA (magenta). Nuclei are visualized by Hoechst 33342 (green). Arrows indicate the mushroom body (MB), antennal lobe (AL), optic lobe (OL), and

Figure 1 continued

subesophageal ganglion (SOG), respectively. Scale bar: 50 µm. (**D**) Representative images of ALs in the adult male brains. Expression patterns of Drice are visualized using SA (magenta). Nuclei are visualized by Hoechst 33342 (green). Scale bar: 50 µm. (**E**) Western blot of biotinylated proteins labeled by Drice::V5::TurboID extracted from adult male brains. Biotinylated proteins are detected by SA. (**F**) A schematic diagram of the TurboID-mediated identification of proximal proteins. Drice-proximal proteins are promiscuously labeled in vivo with the administration of biotin. Then, biotinylated proteins are purified using NeutrAvidin magnetic beads and are subsequently analyzed by mass spectrometry. (**G**) A summary of mass spectrometry analysis of proteins of the adult male brains. Among 643 proteins identified, 158 proteins were detected as highly specific proteins to *Drice::V5::TurboID* flies compared to wild-type flies (FC [*Drice::V5::TurboID/w[1118]*]>10). (**H**) Gene Ontology analysis [Cellular Compartment] of 158 Drice-proximal proteins (FC [*Drice::V5::TurboID/w[1118]*]>10). Cyan: neuronal-related fraction; red: membrane fraction.

The online version of this article includes the following source data and figure supplement(s) for figure 1:

**Source data 1.** Uncropped raw blot images of *Figure 1B and E*.

**Source data 2.** Uncropped annotated blot images of *Figure 1B and E*.

**Figure supplement 1.** Expression patterns of Drice in the adult brain in males and females.

## Drice is proximal to cell membrane proteins rather than cytosolic proteins in the adult brains

To comprehensively identify proximal proteins of Drice, we employed a proximity labeling approach. Specifically, we used *Drice::V5::TurboID* knocked-in flies fed 100 µM biotin. We confirmed biotinylation of Drice-proximal proteins in the adult brains (*Figure 1E*). After purifying the biotinylated proteins using NeutrAvidin magnetic beads, we performed liquid chromatography-tandem mass spectrometry (LC-MS/MS) analysis of the samples (*Figure 1F*). We identified 643 proteins (*Figure 1G*, *Supplementary file 1, table S1*), of which 158 were highly specific to *Drice::V5::TurboID* flies compared to wild-type flies (*Drice::V5::TurboID/w[1118]*>10; *Figure 1G*, *Supplementary file 1, table S1*). Gene Ontology (GO) analysis revealed that Drice-proximal proteins were enriched in the plasma membrane, synaptic vesicles, postsynaptic membranes, and adherens junctions (*Figure 1H*, *Supplementary file 2, table S2*). This finding suggests that Drice is localized primarily in cell membrane compartments rather than in the cytoplasm of the adult brain. Additionally, because the SA signal of *Drice::V5::TurboID* is mainly observed in the AL where the axons of ORNs project (*Figure 1C*), the enrichment of cell membrane proteins in the proximal proteins of Drice is reasonable.

## A specific splicing isoform of Fasciclin 3 is in proximity to Drice

During our analysis of proteins proximal to Drice, we discovered that the expression patterns of Fasciclin 3 (Fas3) in the adult brain closely resembled those of Drice, especially within the OLs, SOGs, and ALs (*Figure 2A and B*). Thus, we focused on Fas3 (*Chiba et al., 1995*), a transmembrane protein containing immunoglobulin-like domains that regulates synaptic target recognition and axon fasciculation (*Kose et al., 1997*). *Fas3* is encoded by seven splicing isoforms, resulting in the production of five protein isoforms that share extracellular domains but differ in intracellular domains with low-complexity sequences (*Figure 2C and D*). Only one specific protein isoform, Fas3 isoform G (Fas3G), was identified in the LC-MS/MS analysis as a Drice-proximal protein in the adult brain (*Figure 1G*, *Supplementary file 1, table S1*). Using AlphaFold2-Multimer (*Evans et al., 2021*; *Mirdita et al., 2022*), Drice was predicted to interact with Fas3A (ipTM = 0.456) and Fas3G (ipTM = 0.372) rather than Fas3C (ipTM = 0.298), Fas3D/E (ipTM = 0.220), and Fas3B/F (ipTM = 0.207). Among Fas3 isoforms, only Fas3G is predicted to interact with Drice with two intracellular regions (*Figure 2—figure supplement 1A*). To validate this result, we generated an antibody specific to Fas3G (*Figure 2C and E*). We confirmed that only the Fas3G was labeled with Drice::V5::TurboID and purified using NeutrAvidin magnetic beads as detected using western blotting (*Figure 2F*). To further confirm the isoform specificity to Drice proximity, we overexpressed 3xFLAG-tagged *Fas3 isoforms A*, *B/F*, *C*, *D/E*, and *G* in ORNs using *Pebbled-Gal4* in a *Drice::V5::TurboID* knocked-in background. After proximity labeling with TurboID, followed by NeutrAvidin purification, we found that Fas3G was highly enriched compared to the other isoforms (*Figure 2G*). These findings suggested that Drice is proximal to a specific isoform of Fas3 in ORNs, regardless of its protein expression level. We also investigated whether Drice and Fas3G statically interact by co-immunoprecipitation. Although Fas3G was labeled by Drice::V5::TurboID, indicating its proximity to Drice within the ORNs of adult brains, Drice did not co-immunoprecipitate with Fas3G (*Figure 2—figure supplement 2A*). This result suggests that while

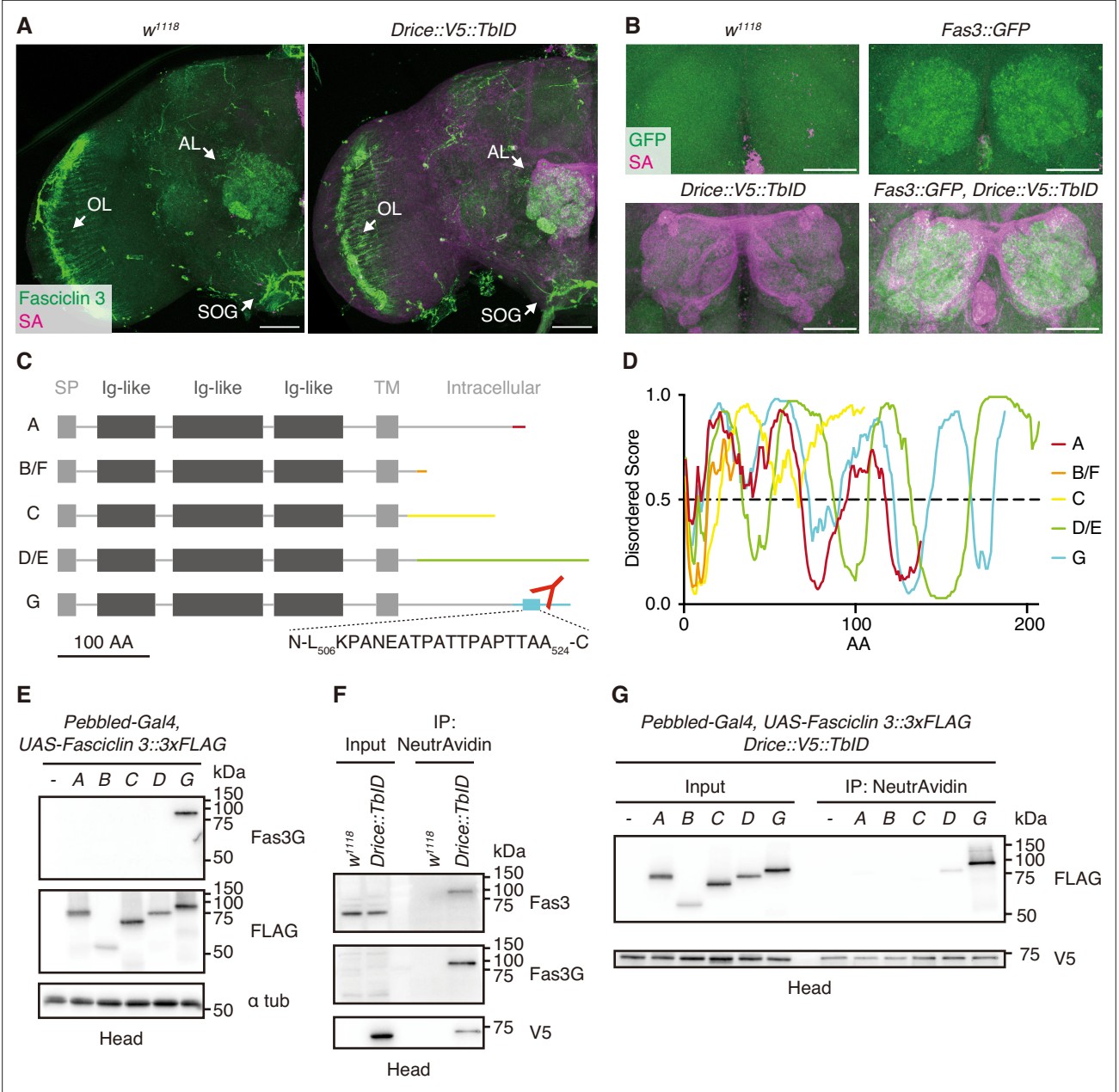

**Figure 2.** A specific isoform of Fasciclin 3 is in proximity to Drice. (**A**) Representative expression patterns of C-terminally V5::TurboID knocked-in tagged Drice (streptavidin [SA]; magenta) and Fasciclin 3 (Fas3, anti-Fasciclin 3 antibody staining; green) in the adult male brains. Arrows indicate the antennal lobe (AL), optic lobe (OL), and subesophageal ganglion (SOG), respectively. Scale bar: 50 µm. (**B**) Representative expression patterns of C-terminally V5::TurboID knocked-in tagged Drice (SA; magenta) and Fas3 (anti-GFP antibody staining; green) in the ALs of the adult male brains. Scale bar: 50 µm. (**C**) Schematic protein structures of Fas3 isoforms. Intracellular regions differ from each other. A peptide region used to raise anti-Fas3G antibody is shown. Signal peptides (SPs), immunoglobulin-like (Ig-like) domains, and transmembrane (TM) domains are shown in boxes. (**D**) IDR predictions of intracellular regions of each protein isoform. Regions wherein the score is more than 0.5 are predicted to be disordered. (**E**) Western blot of proteins extracted from adult heads overexpressing each *3xFLAG-tagged Fas3 isoform*. Anti-Fas3G antibody specifically detects Fas3 isoform G. (**F**) Western blot of endogenous Fas3 expressed in adult male heads. Drice-proximal proteins biotinylated by C-terminally knocked-in tagged V5::TurboID are purified using NeutrAvidin. (**G**) Western blot of 3xFLAG-tagged Fas3 isoforms overexpressed in adult head. Drice-proximal proteins biotinylated by C-terminally knocked-in tagged V5::TurboID are purified using NeutrAvidin.

The online version of this article includes the following source data and figure supplement(s) for figure 2:

**Source data 1.** Uncropped raw blot images of *Figure 2E, F, and G*.

**Source data 2.** Uncropped annotated blot images of *Figure 2E, F, and G*.

*Figure 2 continued on next page*

Figure 2 continued

**Figure supplement 1.** Predicted protein complexes of Drice and each Fas3 isoform.

**Figure supplement 2.** Biochemical property of Fas3s with caspases.

**Figure supplement 2—source data 1.** Uncropped raw blot images of *Figure 2—figure supplement 2A and B*.

**Figure supplement 2—source data 2.** Uncropped annotated blot images of *Figure 2—figure supplement 2A and B*.

Drice and Fas3G are proximate, their interaction is not static, highlighting the utility of proximity labeling as a superior technique to conventional co-immunoprecipitation for identifying proteins that are spatially close.

## Fasciclin 3s are not substrates of caspases

It is often observed that proteins in close proximity to caspases are their preferred substrates (*Shinoda et al., 2023*). To investigate this, we examined whether any of the Fas3 isoforms are substrates for caspase by expressing all isoforms in *Drosophila* S2 cells. However, we found that none of the Fas3 isoforms were cleaved by caspase following the induction of apoptosis (*Figure 2—figure supplement 2B*). Therefore, Fas3s are not substrates of caspase.

## MASCaT: a cell-type-specific highly sensitive caspase activity reporter

We investigated whether Fas3G regulates caspase activity in lethal or non-lethal processes. Caspase-Tracker/CasExpress is a highly sensitive reporter of caspase activity that uses caspase-activatable Gal4 tethered to the membrane by mCD8 (*Ding et al., 2016*; *Tang et al., 2015*). However, reliance on Gal4 to report their activity presents a challenge in reconciling Gal4/UAS-dependent gene manipulations. To overcome this limitation, we created a new reporter to capture caspase activation independent of caspase-activatable Gal4. We replaced the caspase-activatable Gal4 with the caspase-activatable QF2 (*mCD8::DQVD::QF2*), where DQVD is a caspase cleavage sequence, and first expressed directly downstream of *UAS* sequence (*Figure 3—figure supplement 1A*). However, the caspase-activatable QF2 induced mNeonGreen expression under *QUAS* sequence even either with a pan-caspase inhibitor, zVAD-fmk, or an uncleavable DQVA mutant probe (*mCD8::DQVA::QF2*, insensitive to caspase-mediated cleavage which serves as a negative control) (*Figure 3—figure supplement 1B*), potentially because the Gal4/UAS binary system expressed excessive mCD8::DQVD/A::QF2 probes that result in caspase activation-independent induction of QF2/QUAS system. Thus, to reduce the amount of probes expression, we generated *Ubi-FRT-STOP-FRT-mCD8::DQVD::QF2*, simultaneously expressing *Flippase* (*FLP*) under *UAS* to restrict reporter expression to the cells of interest (*Figure 3A*). *Drosophila* S2 cells undergo weak caspase activation only by transfection (*Shinoda et al., 2023*). Taking advantage of that, this system reports caspase activity through the caspase-activatable QF2/QUAS system, expressing mNeonGreen, which is not detected by caspase-non-activatable QF2 (*Figure 3B*). We named this novel reporter a MASCaT (*Figure 3A*). Using MASCaT, we detected caspase activity, specifically in the neurons of adult brains, using *ELAV-Gal4* (*Figure 3C*). Caspase activity was prominent in OLs (*Figure 3C*), which is consistent with the previously reported activation pattern of caspase detected by CaspaseTracker/CasExpress (*Ding et al., 2016*; *Tang et al., 2015*). Previously, we reported that a specific subset of ORNs, including Or42b neurons, undergoes age-associated caspase activation (*Chihara et al., 2014*). Consistent with this finding, we detected age-dependent caspase activation in the AL (*Figure 3C*). We further confirmed the age-dependent caspase activation in Or42b neurons (*Figure 3D*). Therefore, our newly established reporter, MASCaT, enables the highly sensitive detection of caspase activity near the plasma membrane in cells of interest reconciling with Gal4/UAS-dependent gene manipulations.

## Fas3G overexpression activates caspase in a non-lethal manner

To investigate the regulatory role of Fas3 in caspase activity in vivo, we used Or42b neurons as a model, which have previously been shown to undergo age-dependent caspase activation (*Chihara et al., 2014*). Initially, we evaluated the effects of *Dronc* overexpression on apoptosis by quantifying cell body numbers in the antenna in young flies. Dronc induced cell loss while catalytically inactive Dronc (Dronc[CG]) did not, suggesting that Dronc induces apoptosis in a catalytic-dependent manner (*Figure 4A and B*). We then evaluated the effects of the overexpression of the executioner caspases

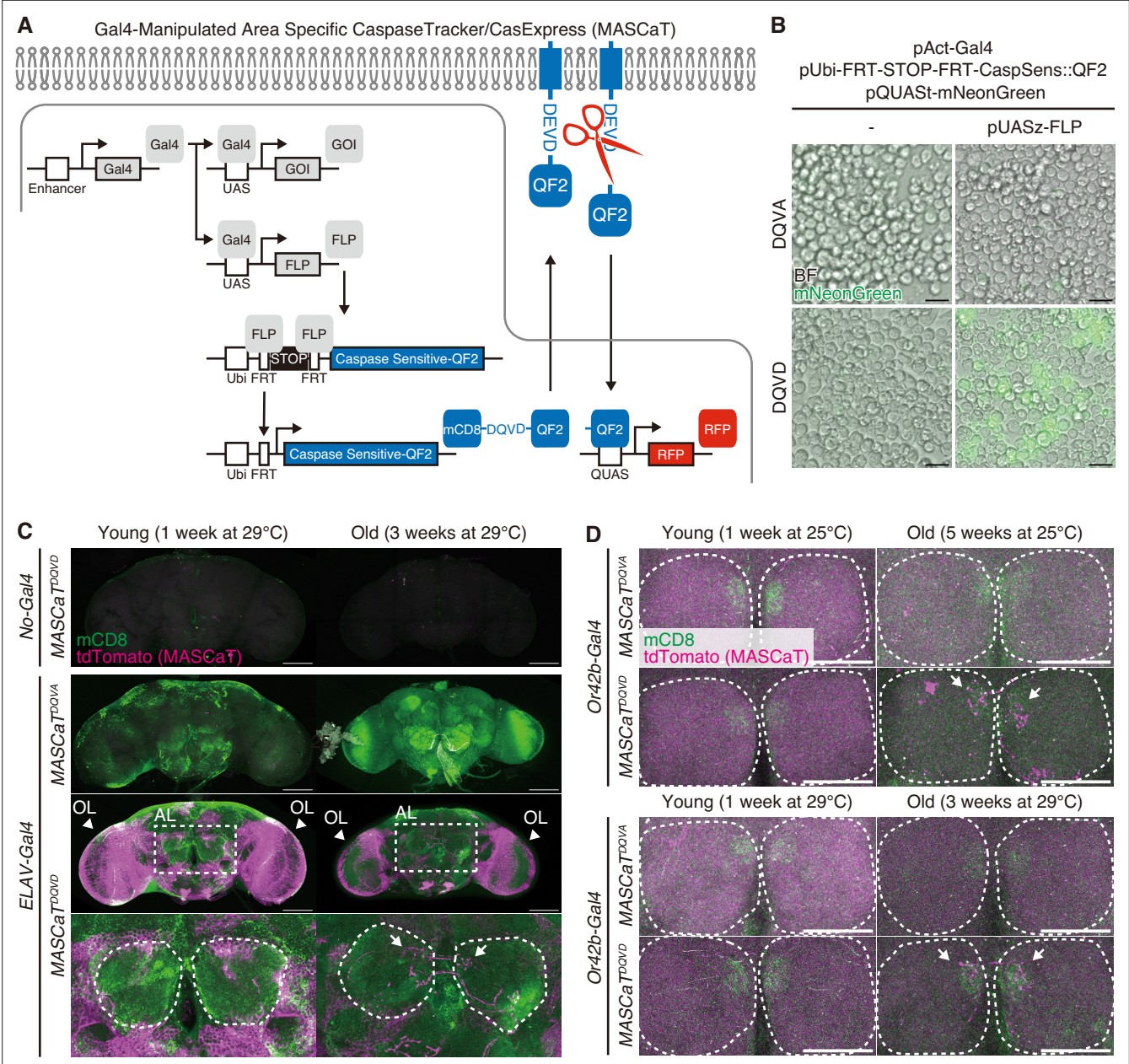

**Figure 3.** A Gal4-Manipulated Area Specific CaspaseTracker/CasExpress (MASCaT). (**A**) A schematic diagram of MASCaT expressed using the Gal4/UAS system with FLP-mediated recombination. Caspase-sensitive-QF2 is activated by caspase-mediated cleavage at the plasma membrane. Cleaved QF2 translocates to the nucleus to induce reporter protein expression downstream of *QUAS* sequence. (**B**) Representative images of *Drosophila* S2 cells expressing caspase-sensitive QF2 probes by the Gal4/UAS system with FLP-mediated recombination. Scale bar: 20 μm. (**C**) Representative images of adult male brains of young (1 week of age) and old (3 weeks of age) flies raised at 29°C expressing MASCaT probe (mCD8; green). Caspase activation is visualized by MASCaT-induced tdTomato (magenta) expression. Arrowheads indicate optic lobes (OLs). Magnified images (white rectangle) show antennal lobes (ALs; white circles). Arrows indicate age-dependent caspase activation at the ALs. Scale bar: 100 μm. (**D**) Representative images of adult male brains of young (1 week of age raised at 25°C or 29°C) and old (5 weeks of age raised at 25°C or 3 weeks of age raised at 29°C) expressing MASCaT probe (mCD8; green). Caspase activation is visualized by MASCaT-induced tdTomato (magenta) expression. White circles show ALs. Arrows indicate age-dependent caspase activation at the ALs. Scale bar: 50 μm.

The online version of this article includes the following figure supplement(s) for figure 3:

**Figure supplement 1.** Overexpression of *mCD8::DQVD::QF2* probes in *Drosophila* S2 cells.

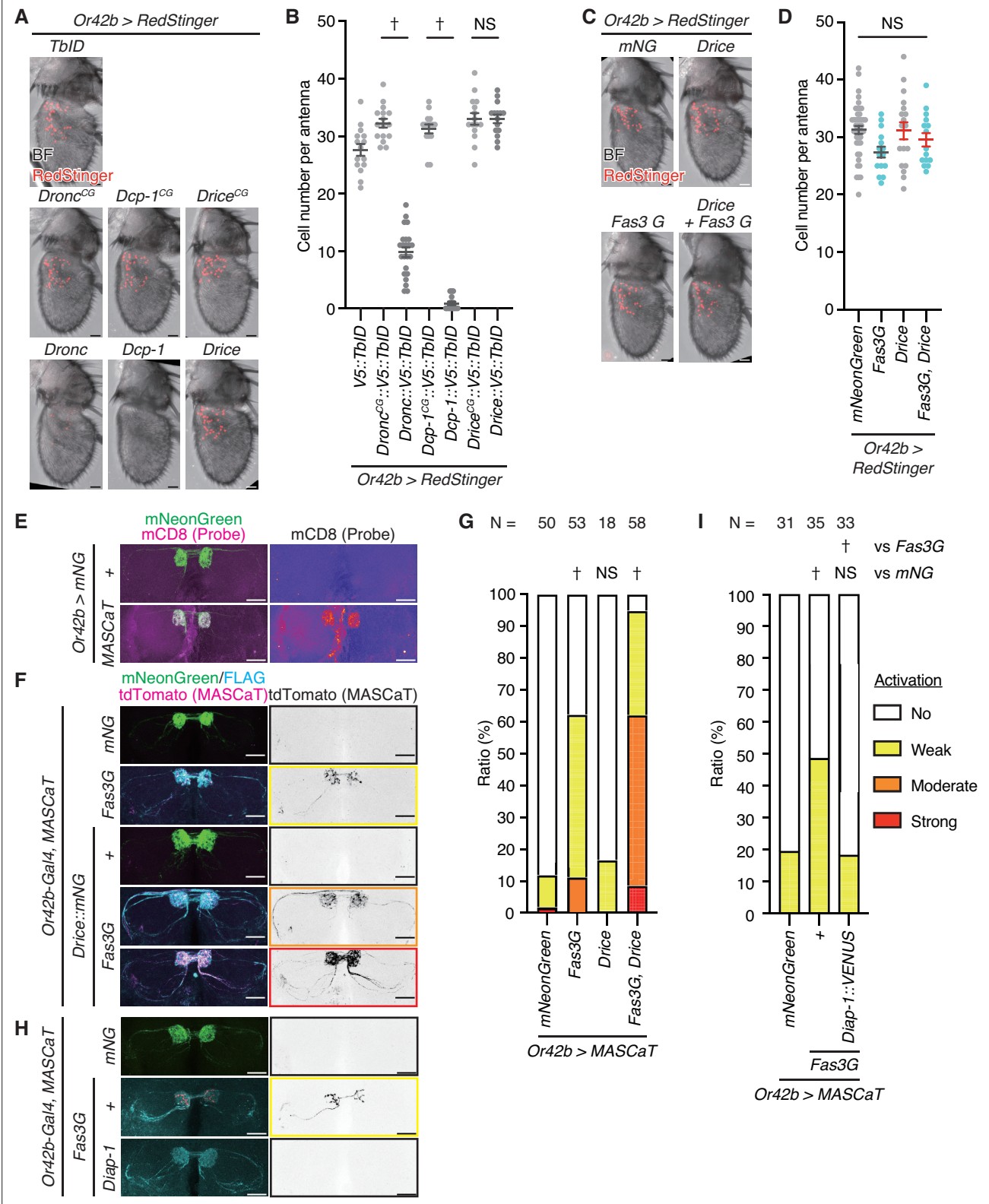

**Figure 4.** *Fasciclin 3 isoform G* overexpression enhances non-lethal caspase activation. (**A**) Representative images of the adult male antennae of 1-week-old flies raised at 29°C. Cell bodies are visualized using RedStinger (red). Scale bar: 20 μm. (**B**) Quantification of cell number of (**A**). Data are presented as mean ± SEM. p-Values were calculated using one-way analysis of variance (ANOVA) with Bonferroni's correction for selected pairs. NS: p>0.05, †: p<0.05. Sample sizes: *V5::TbID* (N=15), *Dronc^CG^:::V5::TbID* (N=16), *Dronc:::V5::TbID* (N=23), *Dcp-1^CG^::V5::TbID* (N=15), *Dcp-1::V5::TbID*

*Figure 4 continued on next page*

*Figure 4 continued*

(N=16), *Drice^CG::V5::TbID* (N=15), *Drice::V5::TbID* (N=15). (**C**) Representative images of the adult male antennae of 1-week-old flies raised at 29°C. Cell bodies are visualized using RedStinger (red). Scale bar: 20 µm. (**D**) Quantification of cell number of (**C**). Data are presented as mean ± SEM. Light blue dots represent *Fas3G* overexpression conditions while red bars represent *Drice* overexpression conditions. p-Values were calculated using one-way ANOVA with Bonferroni's correction for every pair. NS: p>0.05. Sample sizes: *mNeonGreen* (N=44), *Fasciclin 3 isoform G* (*Fas3G*; N=15), *Drice* (N=18), *Fas3G+Drice* (N=15). (**E**) Representative images of the antennal lobes (ALs) of male brains of 1-week-old flies raised at 29°C expressing MASCaT probe (mCD8; magenta). Expression of the probes is restricted to the *Or42b-Gal4* positive (mNeonGreen; green) regions. Scale bar: 30 µm. (**F**) Representative images of the ALs of brains of 1-week-old flies raised at 29°C with caspase activation visualized by MASCaT-induced tdTomato (magenta) expression. Expression of mNeonGreen (green) or 3xFLAG-tagged Fas3G (cyan) is simultaneously visualized. Scale bar: 30 µm. (**G**) Quantifications of caspase activity detected by MASCaT of (**F**). p-Values were calculated using chi-square test with Bonferroni's correction using mNeonGreen as control. NS: p>0.05, †: p<0.05. Sample sizes are shown in the graph. (**H**) Representative images of the ALs of brains of 1-week-old flies raised at 29°C with caspase activation visualized by MASCaT-induced tdTomato (magenta) expression. Expression of mNeonGreen (green) or 3xFLAG-tagged Fas3G (cyan) is simultaneously visualized. Scale bar: 30 µm. (**I**) Quantifications of caspase activity detected by MASCaT of (**H**). p-Values were calculated using chi-square test with Bonferroni's correction using mNeonGreen and Fas3G as controls, respectively. NS: p>0.05, †: p<0.05. Sample sizes are shown in the graph.

The online version of this article includes the following source data and figure supplement(s) for figure 4:

**Source data 1.** Data used for graphs presented in *Figure 4B, D, G, and I*.

**Figure supplement 1.** Loss-of-function analysis of Fas3G.

**Figure supplement 1—source data 1.** Uncropped raw blot images of *Figure 4—figure supplement 1B*.

**Figure supplement 1—source data 2.** Uncropped annotated blot images of *Figure 4—figure supplement 1B*.

**Figure supplement 1—source data 3.** Data used for graphs presented in *Figure 4—figure supplement 1D*.

---

*Dcp-1* and *Drice*. Interestingly, although Dcp-1 induced complete cell loss, Drice did not (***Figure 4A and B***), suggesting that Drice requires an initiator caspase for activation, whereas Dcp-1 does not. These results indicate that Or42b neurons are competent in caspase-mediated cell loss and thus serve as a model for evaluating apoptosis. However, the expression of *Fas3G*, even in combination with *Drice*, did not reduce the number of cells, implying that *Fas3G* overexpression did not induce apoptosis (***Figure 4C and D***).

Subsequently, we assessed the regulation of non-lethal caspase activity by Fas3G in vivo. We first confirmed that MASCaT probes were distributed in the axons of ORNs (***Figure 4E***). Using MASCaT, we observed weak caspase activation even in the control group at a young age (***Figure 4F and G***). The expression of *Fas3G* significantly promoted caspase activation (***Figure 4F and G***), indicating that Fas3G regulates non-lethal caspase activation. Moreover, the simultaneous expression of *Drice* and *Fas3G* significantly enhanced caspase activation (***Figure 4F and G***), signifying the promoting effect of their proximity. Moreover, the concurrent expression of Diap-1, an E3 ligase known to inhibit caspase activation (***Ditzel et al., 2008***; ***Hay et al., 1995***; ***Lee et al., 2011***), mitigated the caspase activation induced by *Fas3G* overexpression (***Figure 4H and I***). This indicates that *Fas3G* overexpression-facilitated non-lethal caspase activation is dependent on the core apoptotic machinery. To test whether endogenous Fas3G regulates caspase activation, we generated a *Fas3G*-specific *shRNA*-expressing strain (***Figure 4—figure supplement 1A and B***). Knockdown of *Fas3* or *Fas3G* in Or42b neurons didn't suppress the inherent, weak caspase activation (***Figure 4—figure supplement 1C and D***). In agreement with this, Drice's presence in the axon of ORNs with Fas3G was observed, yet its presence was not strictly contingent on Fas3 or Fas3G (***Figure 4—figure supplement 1E***). Given the numerous other membrane proteins in proximity to Drice (***Figure 1G and H*** and ***Supplementary file 1, table S1***), it is probable that additional proteins may be responsible for recruiting Drice to the axon. These findings collectively imply that while Fas3G is not essential for mild endogenous caspase activation, overexpression of *Fas3G* does enhance non-lethal caspase activation in ORNs, thus providing a valuable model for exploring the molecular processes that facilitate caspase activation without leading to cell death.

## Fas3G is proximal to Dronc, which is required for non-lethal caspase activation

Next, we investigated the underlying mechanism that enhances non-lethal activation facilitated by *Fas3G* overexpression. Executioner caspases, Drice and Dcp-1, are activated by initiator caspase, Dronc (***Figure 1A***). We precisely analyzed the expression pattern of Dronc in adult brains and found that while its expression was low in control, overexpression of *Fas3G* in ORNs markedly increased

Dronc levels in those neurons (*Figure 5A*). Moreover, akin to Drice, Dronc was also found in close proximity to the overexpressed Fas3G, but not to the overexpressed Fas3C in ORNs (*Figure 5B*). Consequently, we examined whether Dronc is required for the non-lethal caspase activation enhanced by *Fas3G* overexpression. Employing MASCaT, we discovered that both knockdown of *Dronc* and the expression of the *dominant-negative form of Dronc* (*Dronc^{DN}*) hindered the non-lethal caspase activation facilitated by *Fas3G* overexpression (*Figure 5C and D*), indicating a dependence on initiator caspase activity. Importantly, the genetic suppression of Dronc activity did not alter the proximity between Drice and Fas3G (*Figure 5E*), suggesting that the inactive Drice proform is primarily proximal to Fas3G. Solo overexpression of *Dronc* led to cell death in Or42b neurons (*Figure 4A and B*), whereas *Fas3G* overexpression elevated Dronc expression without causing cell death (*Figures 4C, D and 5A*). Collectively, these findings imply that the non-lethal caspase activation driven by *Fas3G* overexpression is orchestrated by an increase in Dronc induced by Fas3G, bringing Dronc in close vicinity to Fas3G and thus enabling caspase activation near the plasma membrane (*Figure 5F*).

## Non-lethal caspase activation regulates innate attraction behavior

Finally, we examined the function of *Fas3G* overexpression-facilitated non-lethal caspase activation. Apple cider vinegar (ACV) excites six glomeruli in the AL, including DM1, which is innervated by the axons of Or42b neurons (*Semmelhack and Wang, 2009*). We have previously reported that Or42b neurons undergo aging-dependent caspase activation and partial cell death, which reduces neuronal activity and attraction behavior in response to ACV (*Chihara et al., 2014*). Inhibition of caspase activity in aged flies restores attraction behavior but does not affect that of young flies (*Chihara et al., 2014*). Using a two-choice preference assay with ACV (*Figure 6A*), we found that 16 hr of starvation combined with 25% ACV in the trap elicited a robust attraction behavior to the vinegar (*Figure 6B*). In contrast, 4 hr of starvation with 1% ACV in the trap resulted in milder attraction behavior, with the preference index value being close to 0 but still showing a positive trend (*Figure 6B*). Under the milder experimental condition, we first confirmed that inhibition of caspase activity through expressing *Dronc^{DN}* didn't affect attraction behavior in young adults (*Figure 6C*), consistent with a previous report (*Chihara et al., 2014*). We then observed that the overexpression of *Fas3G*, which activates caspases, did not impair attraction behavior. Instead, it rather appeared to enhance the tendency for attraction behavior (*Figure 6C*), suggesting that Fas3G promotes attraction behavior. Finally, we found that inhibiting *Fas3G* overexpression-facilitated non-lethal caspase activation by expressing *Dronc^{DN}* strongly promoted attraction to ACV (*Figure 6C*). Overall, these results suggest that *Fas3G* overexpression has a dual function: it enhances attraction behavior while also triggering non-lethal caspase activation, which counteracts the behavioral response, functioning as a regulatory brake without causing cell death.

## Discussion
### Proximal proteins of executioner caspases regulate non-lethal caspase activity

It is important to identify the proximal proteins of executioner caspases, which cleave a wide range of substrates and are essential for apoptosis (*Julien and Wells, 2017*; *Kumar, 2007*), as proximity often dictates substrate specificity (*Shinoda et al., 2023*). In humans, caspase-3 and caspase-7 have been shown to share but also have discrete substrate preferences (*Walsh et al., 2008*). Caspase-7 leverages an exosite to facilitate interaction with RNA and enhance the proteolysis of RNA-binding proteins (*Boucher et al., 2012*; *Desroches and Denault, 2019*). Thus, protein proximity can be used to precisely regulate specific cellular functions by selectively cleaving a limited pool of substrates in non-lethal scenarios compared with lethal cleavages where all the substrates are potentially cleaved during apoptosis. We showed that executioner caspases, before activation as proforms, are exclusively sequestered in specific subcellular domains, including the plasma membrane of axons. We found that *Fas3G* overexpression facilitates non-lethal activation, underscoring the importance of protein proximity not only in substrate cleavage but also in localizing non-lethal activation to the vicinity of the plasma membrane of axons (*Figure 5F*). Our results suggest that subcellularly restricted caspase activation mediated by caspase-proximal proteins is one of the potential mechanisms regulating caspase activation without inducing cell death.

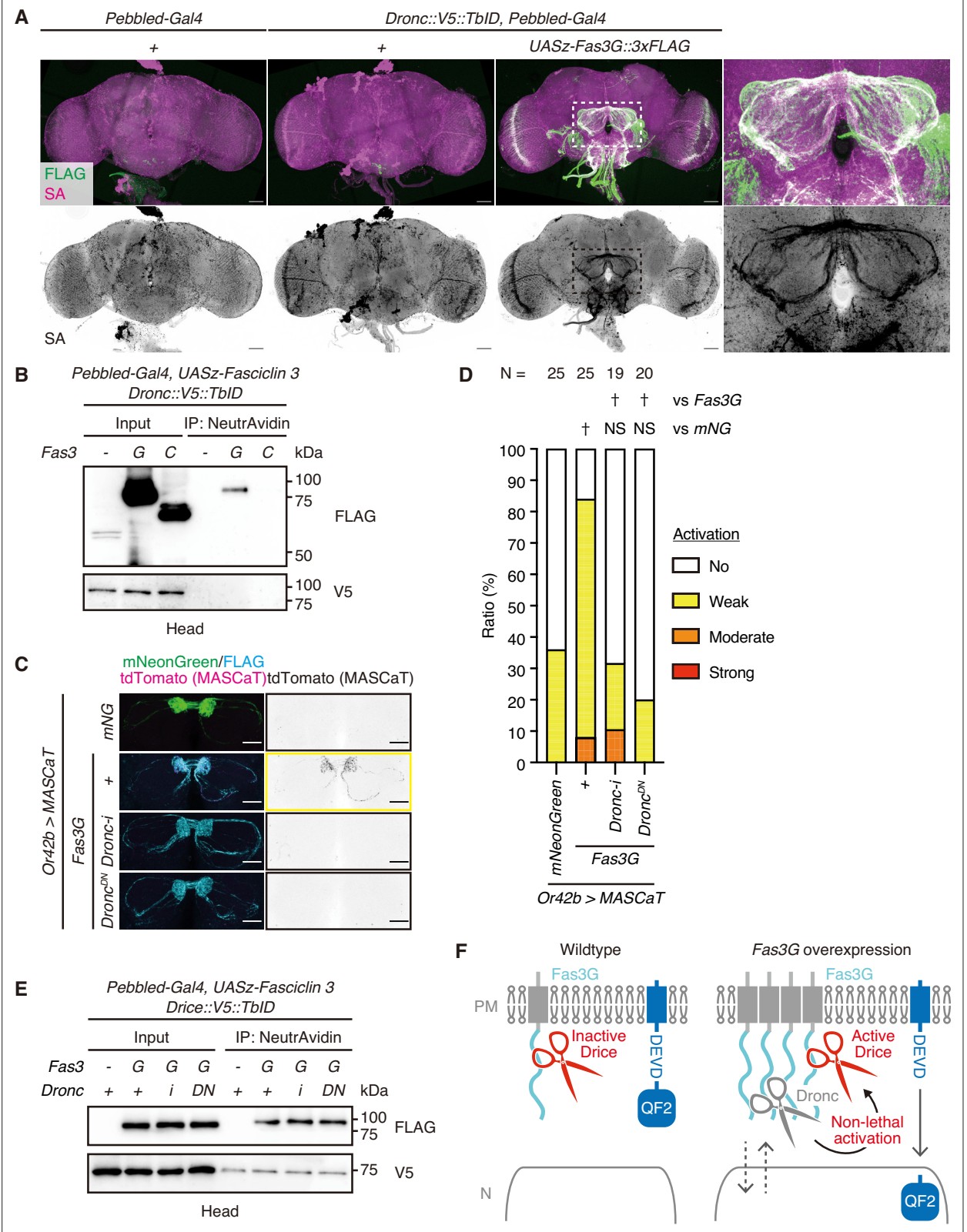

**Figure 5.** *Fasciclin 3 isoform G* overexpression enhances non-lethal caspase activation in a Dronc-dependent manner. (**A**) Representative images of adult male brains. Expression patterns of C-terminally V5::TurboID knocked-in tagged Dronc are visualized using streptavidin (SA; magenta) with or without the expression of 3xFLAG-tagged Fasciclin isoform G (Fas3G; green). Magnified images (white rectangle) show antennal lobes (ALs). Scale bar: 50 μm. (**B**) Western blot of adult male heads. Dronc-proximal proteins biotinylated by C-terminally knocked-in tagged V5::TurboID are purified using

*Figure 5 continued on next page*

*Figure 5 continued*

NeutrAvidin. (**C**) Representative images of the AL of brains of 1-week-old flies raised at 29°C with caspase activation visualized by Gal4-Manipulated Area Specific CaspaseTracker/CasExpress (MASCaT)-induced tdTomato (magenta) expression. Expression of mNeonGreen (green) or 3xFLAG-tagged Fas3G (cyan) is simultaneously visualized. Scale bar: 30 µm. (**D**) Quantifications of caspase activity detected using MASCaT of (**C**). p-Values were calculated using chi-square test with Bonferroni's correction using mNeonGreen and Fas3G as controls, respectively. NS: p>0.05, †: p<0.05. Sample sizes are shown in the graph. (**E**) Western blot of adult male heads. Drice-proximal proteins biotinylated by C-terminally knocked-in tagged V5::TurboID are purified using NeutrAvidin. (**F**) A schematic model of subcellularly restricted non-lethal caspase activation at the plasma membrane (PM). Inactive Drice is in proximity to Fas3G in wild-type condition. *Fas3G* overexpression induces Dronc expression. Then, Dronc comes close to Fas3G and activates Drice. Activated Drice non-lethally cleaves substrate near PM, including MASCaT probes.

The online version of this article includes the following source data for figure 5:

**Source data 1.** Uncropped raw blot images of *Figure 5B and E*.

**Source data 2.** Uncropped annotated blot images of *Figure 5B and E*.

**Source data 3.** Data used for graphs presented in *Figure 5D*.

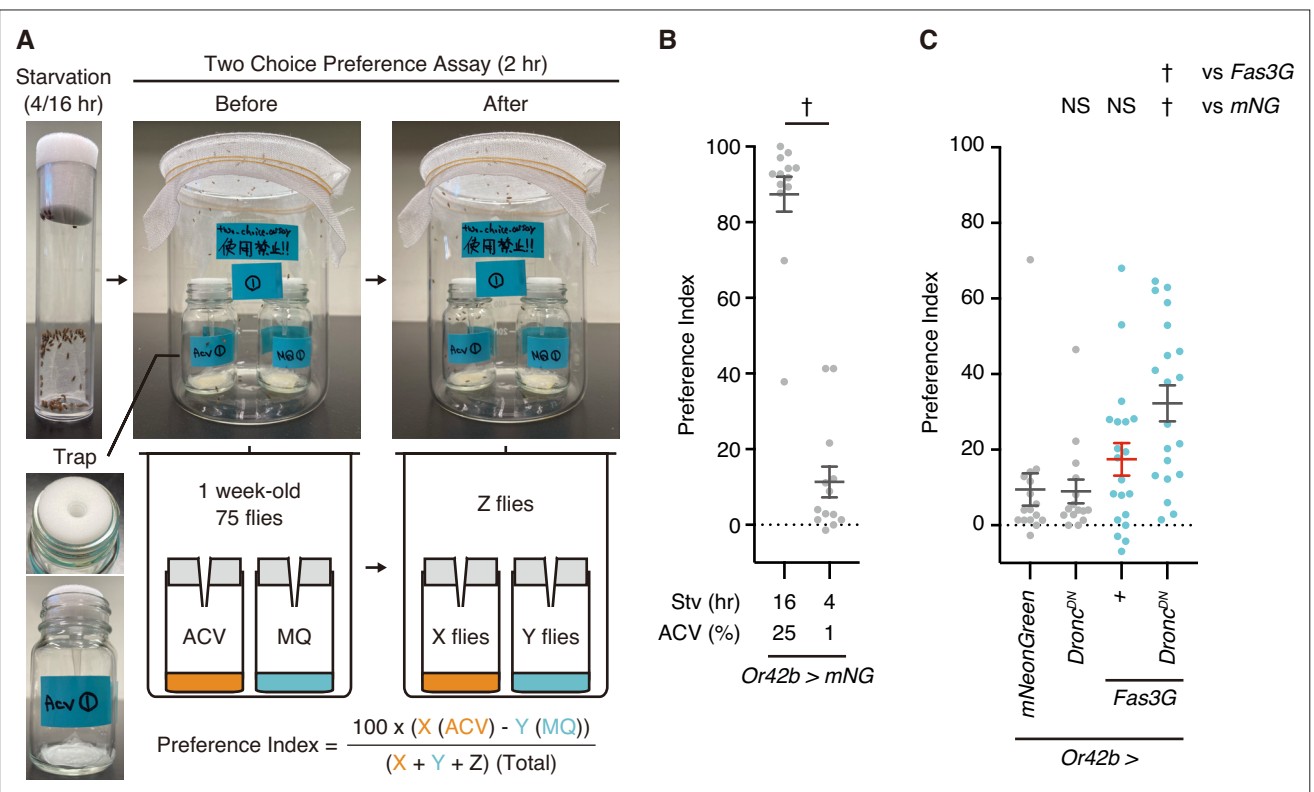

**Figure 6.** *Fasciclin 3 isoform G* overexpression-facilitated caspase activation regulates attraction behavior. (**A**) Images and a schematic diagram of two-choice preference assay. Flies are starved for 4 or 16 hr before the assay. One-week-old young flies are left for 2 hr in the bottle with two traps. After the assay, preference index is calculated by counting the number of flies left in the trap with apple cider vinegar (ACV) as X, in the trap with MilliQ (MQ) water as Y, and other (field) compartments as Z. (**B**) Preference index of 1-week-old male flies raised at 29°C. Flies are starved for 16 or 4 hr and are subsequently tested in response to 25% or 1% ACV in a two-choice preference assay. Data are presented as mean ± SEM. p-Values were calculated using unpaired t-test. †: p<0.05. Sample sizes: 16 hr Stv/25% AC (N=13 [915 flies]), 4 hr Stv/1% ACV (N=13 [945 flies]). (**C**) Preference index of 1-week-old male flies raised at 29°C. Flies are starved for 4 hr and are subsequently tested in response to 1% ACV in a two-choice preference assay. Data are presented as mean ± SEM. Light blue dots represent *Fas3G* overexpression conditions, while a red bar indicates caspase-activated condition. p-Values were calculated using one-way analysis of variance (ANOVA) followed by Dunnets' multiple comparison test using mNeonGreen and Fas3G as controls. NS: p>0.05, †: p<0.05. Sample sizes: mNeonGreen (N=16 [1159 flies]), Dronc$^{DN}$ (N=15 [1099 flies]), Fas3G (N=20 [1419 flies]), Fas3G+Dronc$^{DN}$ (N=20 [1318 flies]).

The online version of this article includes the following source data for figure 6:

**Source data 1.** Data used for graphs presented in *Figure 6B and C*.

## MASCaT as a highly sensitive reporter for identifying non-lethal caspase regulators

Various caspase activity reporters have been developed, including SCAT3, a FRET-based reporter with superior temporal resolution (*Takemoto et al., 2003*), CaspaseTracker/CasExpress, Caspase-activatable Gal4-based reporters with superior sensitivity (*Ding et al., 2016*; *Tang et al., 2015*). However, to date, no reporter has been developed that is sensitive enough to identify non-lethal regulators of caspases in cells of interest. In this study, we developed a novel caspase reporter, MASCaT, that can detect caspase activation with high sensitivity in a cell-type-specific manner following gene manipulation. Using MASCaT, we identified previously overlooked regulators of non-lethal activation of caspases in a cell-type-specific manner. Proximal proteins of executioner caspases are potential non-lethal regulators, and combining TurboID-mediated identification of these proteins will pave the way for identifying non-lethal components and broaden our understanding of the molecular mechanism of non-lethal caspase activation.

## Non-lethal caspase activation regulates neuronal activity and behaviors

We previously reported that age-dependent caspase activation reduces neuronal activity and innate attraction behavior (*Chihara et al., 2014*). Here, we found that, while *Fas3G* overexpression did not impair attraction behavior by itself, caspase activation upon *Fas3G* overexpression suppressed attraction behavior toward ACV (*Figure 6C*). In mice, L1CAM, an immunoglobulin-like cell adhesion molecule similar to Fas3, facilitates neuronal excitability by regulating the voltage-gated Na+ channels (*Valente et al., 2016*). Thus, we hypothesize that non-lethal caspase activation antagonizes the neuronal excitability promoted by Fas3G. In neurons, caspases have diverse non-lethal functions, including axon degeneration and macro- and micro-pruning, to regulate synaptic plasticity (*Dehkordi et al., 2022*; *Mukherjee and Williams, 2017*). In presynaptic terminals, it has been recently reported that caspase-3 modulates synaptic vesicle pools via autophagy (*Gu et al., 2021*). Thus, it is possible that non-lethal activation regulates synaptic pools and neuronal activity in a reversible manner without killing the cells. Analyzing the functions mediated by Drice and Fas3G proximity, including the identification of specific substrates, will pave the way for identifying diverse CDPs and understanding their molecular mechanisms.

While suppressing innate attraction behavior does not seem beneficial by itself, it should be a great advantage in decision-making. To survive, flies must constantly make appropriate behavioral decisions (e.g. feeding versus mating) depending on their physiological conditions. In males, a subset of neurons regulated by tyramine controls feeding versus mating behaviors (*Cheriyamkunnel et al., 2021*). In regard to that, it is important to adequately promote or suppress neuronal activity, which controls the behaviors of the organisms, by precisely controlling caspase activation in a subcellularly restricted manner in neurons. Compared with lethal activation, which irreversibly removed the neurons of interest, suppressing neuronal activity with non-lethal reversible caspase activation is beneficial for decision-making depending on their varying physiological conditions. As regulating caspase-proximal proteins enables non-lethal activation, which is not achieved by directly manipulating caspase expression, our results open the possibility of the methodological development for reversible manipulation of neuronal activity via non-lethal caspase activity.

## Proximal proteins of executioner caspases define the site of caspase activation 'hotspot'

Although the core apoptotic machinery has been well characterized during the past 30 years (*Nakajima and Kuranaga, 2017*; *Shinoda and Miura, 2024*), the precise regulation of CDPs remains incompletely elucidated. In this study, we identified that *Fas3G* overexpression activates Drice. Mechanistically, we found that the *Fas3G* overexpression induced the expression of initiator caspase Dronc, which also comes close to Fas3G, regulating non-lethal caspase activation. Direct overexpression of Dronc alone results in cell death within ORNs (*Figure 4A and B*). However, when Dronc expression is upregulated due to *Fas3G* overexpression, it does not lead to cell death in ORNs (*Figure 4C and D*). The exact process by which Dronc expression is increased in response to *Fas3G* overexpression remains to be elucidated. It is known that Dronc can be upregulated via the ecdysone signaling pathway (*Dorstyn et al., 1999*) and the Hippo signaling pathway (*Verghese et al., 2012*). Therefore, the overexpression of *Fas3G* might potentially activate Dronc expression through one or both signaling pathways. Fas3

is an immunoglobulin-like domain-containing transmembrane protein that regulates axonal fasciculation and neuronal projections (*Chiba et al., 1995*). NCAM, a mammalian homologue of Fascicilin 2, is an immunoglobulin-like domain-containing cell adhesion molecule. NCAM can physically bind to caspase-8, and depending on NCAM clustering, caspase-8 and caspase-3 are activated and contribute to neurite outgrowth by cleaving its substrate spectrin (*Westphal et al., 2010*). Thus, it seems likely that cell adhesion molecule-mediated subcellularly restricted caspase activation is conserved among species. Indeed, caspase-9/Apaf-1-dependent caspase-3 activation, without resulting in cell death, has also been observed in the axon of olfactory sensory neurons during mouse development (*Ohsawa et al., 2010*).

While the restriction of the extent and spread of activated caspases is proposed as one of the potential molecular mechanisms for non-lethal activation (*Mukherjee and Williams, 2017*), experimental evidence is limited. A critical insight from our research is that the executioner caspase is pre-compartmentalized in proximity to a specific set of proteins prior to activation, poised for the initiator caspase to approach. Consequently, the 'hotspot' for non-lethal caspase activity appears to be determined by these neighboring proteins of executioner caspases. While the 'hotspot' located in the axon of the ORN may exert an inhibitory effect on neuronal activity, it could also play a role in avoiding cell death by localizing caspase activity, especially during development.

Recently, a protein interactome analysis in humans showed that tissue-preferentially expressed proteins that interact with known apoptosis regulators participate in apoptosis sensitization (*Luck et al., 2020*). Given that CDPs are often cell- and tissue-type-specific processes, the identification of modulators of context-dependent apoptotic pathways will aid in understanding the precise molecular regulation of CDPs. Importantly, alternative splicing can diversify protein-protein interaction networks. A recent report showed that minor spliceosomes regulate alternative splicing, including Fas3. A specific isoform of Fas3 has distinct functions and cannot be substituted by other isoforms during neuronal development (*Li et al., 2020*). Our results support the idea that alternative splicing may not only diversify the protein itself, but also its interaction networks to regulate module-dependent activation. Further interactome analysis using TurboID will reveal the regulatory mechanisms of the non-lethal activation of caspases.

## Methods

### Fly strains and rearing conditions

We raised the flies in standard *Drosophila* medium at 25°C. For in vivo biotin labeling experiments, flies were raised in standard *Drosophila* medium supplemented with 100 μM (+)-biotin (#029-08713, WAKO) for at least 5 days. The fly strains used in this study are listed in *Supplementary file 3, table S3*. The detailed genotypes corresponding to each figure are provided in *Supplementary file 4, table S4*.

### Protein preparation from *Drosophila* tissues

Adult heads or brains of the given genotype were dissected in phosphate-buffered saline (PBS) and lysed with RIPA buffer (50 mM Tris-HCl, pH 8.0, 150 mM sodium chloride, 0.5 wt/vol% sodium deoxycholate, 0.1% wt/vol sodium dodecyl sulfate, and 1.0% wt/vol NP-40) supplemented with cOmplete ULTRA EDTA-free protease inhibitor cocktail (#05892953001, Roche). Samples were homogenized and centrifuged at 20,000×*g*, 4°C for 10 min. The supernatants were collected and snap-frozen in liquid nitrogen. Protein concentrations were determined using the BCA assay (#297-73101, WAKO) following the manufacturer's protocol. The samples were mixed with 6× Laemmli buffer, boiled at 95°C for 5 min, and then subjected to purification of biotinylated proteins and western blot analysis.

### Western blotting

Each sample was separated using SDS-PAGE. The proteins were then transferred onto Immobilon-P PVDF membranes (#IPVH00010; Millipore) for immunoblotting. Membranes were blocked with 4% skim milk diluted in 1× TBST. Immunoblotting was performed using the antibodies mentioned below diluted in 4% skim milk diluted in 1× TBST. Signals were visualized using Immobilon Western Chemiluminescent HRP Substrate (#WBKLS0500; Millipore) and FUSION SOLO. 7S. EDGE imaging station

(Vilber-Lourmat). Contrast and brightness adjustments were applied equally using Fiji (ImageJ) software (NIH Image).

The primary antibodies used included mouse anti-V5 monoclonal antibody (1:5,000, #46-0705, Invitrogen), mouse anti-FLAG M2 monoclonal antibody (1:5000, Sigma), mouse anti-alpha tubulin (DM1A) monoclonal antibody (1:10,000, #T9026, Sigma), mouse anti-Actin monoclonal antibody (1:5000; #A4700, Sigma), mouse anti-Fasciclin 3 (7G10) antibody (1:50, #AB_528238, DSHB), rabbit anti-Fasciclin 3 isoform G antibody (1:50, this study), mouse anti-*Drosophila* Lamin B monoclonal antibody (1:1000, #ADL67.10, DSHB). The secondary antibodies used included goat anti-rabbit IgG HRP-conjugated antibody (1:5000, #7074S, CST) and goat/rabbit/donkey anti-mouse IgG HRP Conjugate (1:5000, #W402B, Promega). The membranes for SA blotting were blocked with 3% BSA diluted in 1× TBST. SA blotting was performed using SA-horseradish peroxidase (1:10,000, #SA10001; Invitrogen) diluted in 3% BSA diluted in 1× TBST.

## Co-immunoprecipitation

Adult heads were lysed on ice using *n*-dodecyl-β-D-maltoside (DDM) IP lysis buffer (50 mM Tris-HCl [pH 7.5], 150 mM NaCl, 0.2% DDM [#341-06161, WAKO]) containing cOmplete ULTRA EDTA-free protease inhibitor cocktail. The samples were adjusted to 150 µg protein/150 µL DDM IP lysis buffer. The samples were incubated overnight at 4°C with 10 µL anti-FLAG M2 Magnetic Beads (#M8823-1ML, Millipore) equilibrated with the DDM IP lysis buffer. The beads were washed three times in wash buffer (50 mM Tris-HCl [pH 7.5], 150 mM NaCl, 0.1% DDM) and boiled for 5 min at 95°C with 50 µL 1× Laemmli buffer. The samples were magnetically separated, and the supernatants were subjected to SDS-PAGE.

## Immunohistochemistry

Adult brains were dissected in PBS and fixed in 4% paraformaldehyde (PFA) at room temperature for 30 min. Adult brains were washed three times with 0.3% PBST (Triton X-100) for 10 min. After blocking with 5.0% normal donkey serum (#S30, Millipore)/PBST (PBSTn) for 30 min at room temperature, tissues were incubated with primary antibody/PBSTn overnight at 4°C. The samples were then washed with PBST three times for 10 min each and incubated with secondary antibody/PBSTn for 2 hr at room temperature. After incubation, the tissues were washed with PBST three times for 10 min. SA and Hoechst 33342 were added during secondary antibody incubation. The samples were mounted on glass slides with SlowFade Gold antifade reagent (#S36939, Invitrogen). Images were captured using a Leica TCS SP8 microscope (Leica Microsystems). Images were analyzed and edited using the Fiji (ImageJ) software. For MASCaT analysis, newly eclosed adults were raised at 29°C for 1 week. Classification of the MASCaT signal was performed by optically assessing the tdTomato intensity. For cell number analysis, newly eclosed adults were raised at 29°C for 1 week. The antennae were dissected as previously described (*Karim et al., 2014*), fixed with 4% PFA at room temperature for 30 min, and washed with 0.4% PBST (Triton X-100) three times for 10 min. Quantification of cell numbers in the antenna was performed using Fiji (ImageJ) software using '3D object counter' for RedStinger positive cells.

The primary antibodies used were mouse anti-Fasciclin 3 (7G10) antibody (1:20, #AB_528238, DSHB), rat anti-GFP (GF090R) monoclonal antibody (1:100, #04404-26, nacalai Tesque), mouse anti-FLAG M2 monoclonal antibody (1:500, F1804, Sigma), and rat anti-mCD8 monoclonal antibody (1:50, #MCD0800, Invitrogen). The secondary antibodies used included Alexa Fluor 488-conjugated donkey anti-mouse IgG antibody (1:500, #A-21202, Thermo Fisher Scientific), Alexa Fluor 488-conjugated donkey anti-rat IgG antibody (1:500, #A-21208, Thermo Fisher Scientific), Alexa Fluor 647-conjugated donkey anti-mouse IgG antibody (1:500, #A-31571, Thermo Fisher Scientific), and Alexa Fluor 633-conjugated goat anti-rat IgG antibody (1:500, #A-21094, Thermo Fisher Scientific). The nuclei were stained with Hoechst 33342 (8 µM, #H3570, Invitrogen). Streptavidin-Cy2 (1:500, #016-220-084, Jackson ImmunoResearch) and Streptavidin-Cy5 (1:500, #016-170-084, Jackson ImmunoResearch) were used to stain the biotinylated proteins.

## Experimental design for LC-MS/MS analyses

Two biological replicates were analyzed for each experimental condition to determine the relative abundance of *Drice::V5::TurboID/w^1118^*.

## Purification of biotinylated proteins

Samples were prepared as previously described (*Shinoda et al., 2023*). Briefly, 100 µg of biotinylated protein-containing lysate (from 100 brains or 20 heads) was subjected to FG-NeutrAvidin beads (#TAS8848 N1171, Tamagawa Seiki) purification. The FG-NeutrAvidin beads (25 µL, approximately 500 µg) were washed three times with RIPA buffer. Benzonase-treated biotinylated protein samples suspended in 1 mL RIPA buffer were incubated overnight at 4°C. Conjugated beads were magnetically isolated and washed with 500 µL of an ice-cold RIPA buffer solution, 1 M KCl solution, 0.1 M $Na_2CO_3$ solution, and 4 M urea solution. For western blot analyses, the purified samples were mixed with Laemmli buffer, boiled at 95°C for 5 min, and then subjected to western blot analysis. For LC-MS/MS analysis, the purified samples were washed with 500 µL ultrapure water (#214-01301, WAKO) and 500 µL of 50 mM ammonium bicarbonate (AMBC). The samples were then mixed with 50 µL of 0.1% Rapigest diluted in 50 mM AMBC, and 5 µL of 50 mM TCEP was subsequently added. Samples were incubated at 60°C for 5 min, and then 2.5 µL 200 mM MMTS was added. One microgram sequence-grade modified trypsin (#V5111, Promega) was then added for on-bead trypsin digestion at 37°C for 16 hr. The beads were magnetically isolated, and 60 µL of the supernatant was collected. Then, 3 µL 10% TFA was added to the supernatants, and the samples were incubated at 37°C for 60 min with gentle agitation. The samples were then centrifuged at 20,000×g, 4°C for 10 min. The peptides were desalinated and purified using a GL-tip SDB (#7820-11200, GL Sciences) following the manufacturer's protocol. The samples were speed-backed at 45°C for 30 min and dissolved in 25 µL of 0.1% formic acid. The samples were then centrifuged at 20,000×g, 4°C for 10 min, and the supernatants were collected. The peptide concentrations were determined using the BCA assay. Finally, 500 ng of the purified protein was subjected to LC-MS/MS analysis.

## LC-MS/MS analyses

LC-MS/MS analyses were performed as previously described (*Shinoda et al., 2023*). Briefly, samples were loaded onto Acclaim PepMap 100 C18 column (75 µm × 2 cm, 3 µm particle size, and 100 Å pore size; #164946, Thermo Fisher Scientific) and separated on nano-capillary C18 column (75 µm × 12.5 cm, 3 µm particle size, #NTCC-360/75-3-125, Nikkyo Technos) using EASY-nLC 1200 system (Thermo Fisher Scientific). The elution conditions are listed in *Supplementary file 5, table S5*. The separated peptides were analyzed using QExactive (Thermo Fisher Scientific) in data-dependent MS/MS mode. The parameters for the MS/MS analysis are listed in *Supplementary file 6, table S6*. The collected data were analyzed using Proteome Discoverer (PD) 2.2 software with the Sequest HT search engine. The parameters for the PD 2.2 analysis are described in *Supplementary file 7, table S7*. Peptides were filtered at a false discovery rate of 0.01 using the Percolator node. Label-free quantification was performed based on the intensities of the precursor ions using a precursor-ion quantifier node. Normalization was performed using the total amount of peptides in all average scaling modes. Proteins with 10 folds or higher abundance relative to the control ($w^{1118}$) were considered for further analysis. MS proteomic data were deposited in the ProteomeXchange Consortium via the jPOST partner repository with the dataset identifier PXD042922 (*Okuda et al., 2017*).

## GO analysis

GO analysis of cellular components was performed using DAVID (*Huang et al., 2009*).

## Generation of rabbit polyclonal anti-Fasciclin 3 isoform G antibody

The peptide corresponding to the 506–524 AA (N-LKPANEATPATTPAPTTAA-C) of Fas3G was used by Eurofins Inc to immunize rabbits to raise a polyclonal antibody. The antibody was affinity-purified using the same peptide provided by Eurofins, Inc.

## IDR prediction

Amino acid sequences were obtained from the FlyBase database (https://flybase.org/). Disordered intracellular regions of Fas3 isoforms were predicted using the PSIPRED protein sequence analysis workbench in DISOPRED3 (http://bioinf.cs.ucl.ac.uk/psipred/).

## Protein complexes prediction with AlphaFold2-Multimer

Amino acid sequences were obtained from the FlyBase database (https://flybase.org/). Protein complexes were predicted using ColabFold v1.5.5: AlphaFold2 using MMseqs2 (*Mirdita et al., 2022*) with the following settings: template_mode: none, msa_mode: mmseqs2_uniref_env, pair_mode: unpaired_paired, model_type: alphafold2_multimer_v3, num_recycles: 3. PAE plots of the structural models ranked first among five models were selected as representatives.

## Molecular cloning

For all isoforms (A, B/F, C, D/E, G) of *pUASz-Fasciclin 3::3xFLAG*, the coding sequences of Fasciclin 3s were PCR-amplified using 3xFLAG-tag harboring primers from *Drosophila* cDNA and were ligated into the *Bam*HI/*Kpn*I-digested *pUASz1.0* vector (#1431, Drosophila Genomics Resource Center [DGRC]) (*DeLuca and Spradling, 2018*) using In-Fusion (#Z9648N, TaKaRa). For all isoforms of *pAc5-Fasciclin 3::3xFLAG*, the coding sequence of *Fasciclin 3s::3xFLAG* was PCR-amplified from corresponding *pUASz-Fasciclin 3::3xFLAG* vectors and was ligated into the *Eco*RI/*Xho*I-digested *pAc5-STABLE2-neo* vector (#32426, Addgene) (*González et al., 2011*) using NEBuilder HiFi DNA Assembly (#E2621L, NEB). For *pUASz-3xFLAG::mNeonGreen*, the coding sequence of mNeonGreen was PCR-amplified using 3xFLAG-tag harboring primers from *pAc5-V5::mNeonGreen* (*Shinoda et al., 2023*) and ligated into the *Xho*I-digested *pUASz1.0* vector using In-Fusion. For *pUASz-Drice::myc::mNeonGreen*, the coding sequence of mNeonGreen was PCR-amplified from *pAc5-V5::mNeonGreen* using myc-tag-harboring primers. The coding sequence of Drice was PCR-amplified from *Drosophila* cDNA. The two fragments were then ligated into the *Xho*I-digested *pUASz1.0* vector using In-Fusion. For *pUASz-V5::TurboID* and *pUASz-Caspase::V5::TurboID*, the coding sequences of *Dcp-1*, *Drice,* and *Dronc* were PCR-amplified from *Drosophila* cDNA. The coding sequence of V5::TurboID was PCR-amplified from *V5-TurboID-NES_pcDNA3* (#107169; Addgene). The two fragments were then ligated into the *Xho*I-digested *pUASz1.0* vector using In-Fusion. Catalytic dead mutations (Dcp-1: C196G [TGC to GGC], Drice: C211G [TGC to GGC], and Dronc: C318G [TGC to GGC]) were introduced by PCR-based site-directed mutagenesis, and the corresponding fragments were ligated into the *Xho*I-digested *pUASz1.0* vector using In-Fusion. For *pJFRC-mCD8::DQVD/A::QF2*, the coding sequence of *mCD8::DQVD/A* was PCR-amplified from genomic DNA of *Caspase-Sensitive/Insensitive-Gal4* (*Tang et al., 2015*). The coding sequence of QF2 was PCR-amplified from *pBPQU* vector (*Kashio et al., 2016*). These fragments were ligated into the *Xho*I/*Xba*I-digested *pJFRC-28K* vector (a gift from H Kazama) using In-Fusion. For *pUbi-FRT-STOP-FRT-mCD8::DQVD/A::QF2*, the FRT-STOP-FRT cassette sequence was PCR-amplified from *pJFRC201-10xUAS-FRT-STOP-FRT-myr::smGFP-HA* (#63166; Addgene). The coding sequence of mCD8::DQVD/A was PCR-amplified from the genomic DNA of *Caspase-Sensitive/Insensitive-Gal4* (*Tang et al., 2015*). The coding sequence of QF2 was PCR-amplified from the *pBPQUw* plasmid (*Kashio et al., 2016*). The three fragments were ligated into *Eco*RI-digested *pUbi83* vector (a gift from D Umetsu) using In-Fusion. For *pQUAST-mNeonGreen*, the coding sequence of mNeonGreen was PCR-amplified from *pAc5-V5::mNeonGreen* and ligated into the *Eco*RI-digested *pQUAST* (#24349, Addgene) vector using In-Fusion. For *pUASz-FLPo*, the coding sequences of FLPo were PCR-amplified from *pQUAST-FLPo* (#24357, Addgene) and ligated into the *Xho*I-digested *pUASz1.0* vector using In-Fusion. For *pWALIUM20-Fas3G-shRNA*, the top and bottom oligonucleotides including *mRNA* sequence specific to isoform G (5'-AATGAACCAAAGCAAGACAAA-3') were annealed in annealing buffer (10 mM Tris-HCl, 0.1 M NaCl, 1 mM EDTA). The annealed fragment was ligated into *Nhe*I/*Eco*RI-digested *pWALIUM20* (#1472, DGRC) vector using DNA Ligation Kit <Mighty Mix> (#6023, TaKaRa).

All established plasmids were sequenced by Eurofins, Inc. The detailed plasmid DNA sequences are available upon request. The established plasmids were injected into $y^1$, $w^{67c23}$; *P[CaryP]attP40* (*pUASz-3xFLAG::mNeonGreen*, *pUASz-Drice::myc::mNeonGreen*, *pUbi-FRT-STOP-FRT-mCD8::DQVD/A::QF2* and *pWALIUM20-Fas3G-shRNA*) or $y^1$, $w^{67c23}$; *P[CaryP]attP2* (*pUASz-Fasciclin 3s::3xFLAG*, *pUASz-V5::TurboID* and *pUASz-Caspase::V5::TurboID*) for PhiC31-mediated transgenesis (BestGene Inc). Each red-eye-positive transformant was isogenized. Integration of the attP landing site was confirmed using genomic PCR.

## Cell lines

*Drosophila* S2 cells (#RCB1153, RIKEN BRC) were grown at 25°C in Schneider's *Drosophila* medium (#21720001, GIBCO) supplemented with 10% (vol/vol) heat-inactivated fetal bovine serum, 100 U/mL penicillin, and 100 μg/mL streptomycin (#168-23191, WAKO). Mycoplasma contamination was assessed using the MycoStrip (#rep-mys-10, InvivoGen) and found to be negative. Cells were seeded in 24-well plates and transfected with the desired plasmids using Effectene Transfection Reagent (#301427, QIAGEN) following the manufacturer's protocol. Sixteen hours after transfection, the cells were reseeded into CELL CULTURE DISH, PS, 35/10 MM (#627870, Grainer BIO-ONE) and incubated for 24 hr. Images were obtained using a Leica AF6000 DMI6000 B microscope (Leica Microsystems). Images were analyzed and edited using Fiji software (NIH Image).

For the substrate cleavage assay, cells were seeded in 12-well plates and were transfected with desired *pAc5-Fas3::3xFLAG* plasmids using Effectene Transfection Reagent following the manufacturer's protocol. After 40 hr of incubation, the cells were treated with 10 μg/mL cycloheximide (CHX, #C7698, Sigma) for apoptosis induction. After 6 hr of incubation, cells were collected using RIPA buffer supplemented with cOmplete ULTRA EDTA-free protease inhibitor cocktail depending on the cell status. For apoptotic cell lysate collection, cells were resuspended by pipetting and were collected into a 1.5 mL tube. The cells were then centrifuged at $800 \times g$, 4°C for 5 min, resuspended in PBS, and centrifuged again using the same parameters. The cell pellets were then lysed with 100 μL RIPA buffer. For the other experimental conditions, the cells were washed thrice with PBS and directly lysed on the plate with 100 μL RIPA buffer. The supernatants were collected and snap-frozen in liquid nitrogen. Protein concentrations were determined using the BCA assay following the manufacturer's protocol. The samples were mixed with 6× Laemmli buffer, boiled at 95°C for 5 min, and then subjected to western blot analysis.

For the validation of Fas3G-shRNA, cells were seeded in 12-well plates and were co-transfected with *pActin-Gal4,* desired *pUASz-Fas3::3xFLAG,* and *pWALIUM20-empty* or *pWALIUM20-Fas3G-shRNA* plasmids using Effectene Transfection Reagent following the manufacturer's protocol. After 40 hr of incubation, the cells were washed thrice with PBS and directly lysed on the plate with 100 μL RIPA buffer. The supernatants were collected and snap-frozen in liquid nitrogen. Protein concentrations were determined using the BCA assay following the manufacturer's protocol. The samples were mixed with 6× Laemmli buffer, boiled at 95°C for 5 min, and then subjected to western blot analysis.

## Aging experiment

Adult male flies were raised in vials containing 15 flies/vial and incubated at 29°C, 65% humidity with 12 hr/12 hr light/dark cycle. The flies were transferred every 2–4 days to vials containing fresh food.

## Two-choice preference assay

Newly eclosed adult male flies were raised in vials containing 25 flies/vial, and incubated at 29°C, 65% humidity with 12 hr/12 hr light/dark cycle. The flies were transferred every 2–4 days to vials containing fresh food for 1 week. The 1-week-old young flies were starved in MilliQ (MQ) water for 4 hr or 16 hr prior to the assay in constant dark condition. About 75 flies were put in a 1000 mL glass beaker, which contained two standards bottles (traps) filled with 500 μL 1% or 25% ACV (Mizkan) and 500 μL MQ water. The bottles were then covered with sponge plugs containing pipette tips. The pipette tip was cut to increase the size of the opening such that only one fly could climb into the bottle at a time. The glass beaker was covered with a mesh fabric (*Figure 6A*). The beakers were set in an incubator at 29°C for 2 hr for the assay in constant dark condition. The preference index equaled the number of flies in the ACV minus the number of flies in the MQ water, divided by the number of total live flies in both inside and outside of the bottles, and multiplied by 100. Values greater than zero indicate a preference for ACV, and values less than zero indicate aversion. We repeated the same experiment with different sets of flies, and the number of times is indicated by N.

## Statistical analysis

Statistical analyses were performed using the GraphPad Prism 8 software (GraphPad). Data are shown as the means ± standard error of the mean. p-Values were calculated using a one-way analysis of variance (ANOVA) with Bonferroni's correction (*Figure 4B* for selected pairs, *Figure 4D* for every pair). p-Values were calculated using the chi-square test with Bonferroni's correction, with mNeonGreen as

a control (*Figure 4G*), LacZ-RNAi as a control (*Figure 4—figure supplement 1D*), and mNeonGreen and Fas3G as controls (*Figures 4I and 5D*). p-Value was calculated using unpaired t-test (*Figure 6B*). p-Values were calculated using one-way ANOVA with Dunnett's multiple comparison test (*Figure 6C*, mNeonGreen and Fas3G as controls). NS: $p > 0.05$, †: $p < 0.05$.

## Materials availability statement

All materials generated in this study, including fly lines, plasmids, a peptide, and an antibody, are available from the corresponding authors upon reasonable request.

## Acknowledgements

We would like to thank Dr. D Umetsu, Dr. H Kazama, Addgene, and DGRC for providing plasmids, Dr. Y Nitta, Dr. A Sugie, and Dr. T Ichinose for helpful discussions, and the Bloomington *Drosophila* Stock Center, Vienna *Drosophila* Resource Center, and Kyoto Stock Center for providing the fly strains. We thank Dr. S Hirayama, Dr. S Murata, and the one-stop sharing faculty center for future drug discovery at the Graduate School of Pharmaceutical Sciences, University of Tokyo, for the LC-MS/MS analysis. We thank Miura's lab members for their technical assistance and discussions; in particular, K Takenaga for preparing the fly food and R Takamoto for experimental support. The MASCaT was established by MASaya Muramoto with critical advice from Shu MASuda at MASayuki Miura lab. We would like to thank Editage (https://www.editage.com) for English language editing of this manuscript. This work was supported by grants from the Ministry of Education, Culture, Sports, Science, and Technology of Japan (KAKENHI Grant Numbers 19K16137, 21K15080, 22H05586, and 23K05747 to NS and 21H04774, 23H04766, 24H00567, and 25H01842 to MMi), the Japan Agency for Medical Research and Development (AMED; Grant number JP21gm5010001 to MMi), and grants from the Takeda Science Foundation and Sumitomo Foundation to NS.

## Additional information

### Funding

| Funder | Grant reference number | Author |
| --- | --- | --- |
| Ministry of Education, Culture, Sports, Science and Technology | 19K16137 | Natsuki Shinoda |
| Ministry of Education, Culture, Sports, Science and Technology | 21H04774 | Masayuki Miura |
| Japan Agency for Medical Research and Development | JP21gm5010001 | Masayuki Miura |
| Takeda Science Foundation | | Natsuki Shinoda |
| Sumitomo Foundation | | Natsuki Shinoda |
| Ministry of Education, Culture, Sports, Science and Technology | 21K15080 | Natsuki Shinoda |
| Ministry of Education, Culture, Sports, Science and Technology | 22H05586 | Natsuki Shinoda |
| Ministry of Education, Culture, Sports, Science and Technology | 23K05747 | Natsuki Shinoda |
| Ministry of Education, Culture, Sports, Science and Technology | 23H04766 | Masayuki Miura |

| Funder | Grant reference number | Author |
|---|---|---|
| Ministry of Education, Culture, Sports, Science and Technology | 24H00567 | Masayuki Miura |
| Ministry of Education, Culture, Sports, Science and Technology | 25H01842 | Masayuki Miura |

The funders had no role in study design, data collection and interpretation, or the decision to submit the work for publication.

## Author contributions

Masaya Muramoto, Nozomi Hanawa, Formal analysis, Investigation, Methodology, Writing – review and editing; Misako Okumura, Takahiro Chihara, Methodology, Writing – review and editing; Masayuki Miura, Conceptualization, Supervision, Funding acquisition, Writing – original draft, Project administration, Writing – review and editing; Natsuki Shinoda, Conceptualization, Formal analysis, Supervision, Funding acquisition, Investigation, Visualization, Methodology, Writing – original draft, Project administration, Writing – review and editing

## Author ORCIDs

Nozomi Hanawa ⬥ http://orcid.org/0009-0001-3852-7508
Misako Okumura ⬥ http://orcid.org/0000-0003-3162-0416
Takahiro Chihara ⬥ https://orcid.org/0000-0001-9989-3619
Masayuki Miura ⬥ http://orcid.org/0000-0001-7444-5705
Natsuki Shinoda ⬥ https://orcid.org/0000-0002-6668-8747

Reviewer #1 (Public review): https://doi.org/10.7554/eLife.99650.3.sa1
Reviewer #2 (Public review): https://doi.org/10.7554/eLife.99650.3.sa2
Author response https://doi.org/10.7554/eLife.99650.3.sa3

# Additional files

## Supplementary files

Supplementary file 1. Proximal protein list. A list of the identified proximal proteins (Abundance ratio (Drice::V5::TurboID/Control) > 10). The UniProt Accessions, Gene Description, and Abundance Ratio (Drice::V5::TurboID/Control) are listed.

Supplementary file 2. Gene Ontology (GO) analysis. The result of GO analysis of Cellular Compartment for Drice-proximal proteins. GO term, p-values, Fold Enrichment, Count, and Genes are listed.

Supplementary file 3. Fly stock list. A list of the fly strains. Genotype, Map, Source or Reference, and Stock# are listed.

Supplementary file 4. Detailed genotypes. Detailed genotypes and their related figure # are presented.

Supplementary file 5. Liquid chromatography (LC) settings. Gradient settings for LC analysis are presented.

Supplementary file 6. Tandem mass spectrometry (MS/MS) settings. Settings used for the MS/MS analysis are presented.

Supplementary file 7. Proteome Discoverer 2.2 settings. Proteome Discoverer 2.2 settings for proteomics analysis are presented.

MDAR checklist

## Data availability

The data that supports the findings of this study are available in the manuscript and supporting files of this article. Source data files have been provided for Figure 1, Figure 2, Figure 2 - figure supplement 2, Figure 4, Figure 4 - figure supplement 1, Figure 5 and Figure 6. The mass spectrometry proteomics data that support the findings of this study have been deposited to the ProteomeXchange

Consortium [http://proteomecentral.proteomexchange.org] via the jPOST partner repository with the dataset identifier PXD042922.

The following dataset was generated:

| Author(s) | Year | Dataset title | Dataset URL | Database and Identifier |
|---|---|---|---|---|
| Shinoda N | 2024 | TurboID-mediated proximal labeling of *Drosophila* caspase Drice in adult brains | https://proteomecentral.proteomexchange.org/cgi/GetDataset?ID=PXD042922 | ProteomeXchange, PXD042922 |

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

## Appendix 1

### Appendix 1—key resources table

| Reagent type (species) or resource | Designation | Source or reference | Identifiers | Additional information |
|---|---|---|---|---|
| Gene (*Drosophila melanogaster*) | Fas3-PA | FlyBase | FLYB: FBpp0080604 | |
| Gene (*Drosophila melanogaster*) | Fas3-PB | FlyBase | FLYB: FBpp0080605 | |
| Gene (*Drosophila melanogaster*) | Fas3-PC | FlyBase | FLYB: FBpp0111716 | |
| Gene (*Drosophila melanogaster*) | Fas3-PD | FlyBase | FLYB: FBpp0111717 | |
| Gene (*Drosophila melanogaster*) | Fas3-PE | FlyBase | FLYB: FBpp0290605 | |
| Gene (*Drosophila melanogaster*) | Fas3-PF | FlyBase | FLYB: FBpp0309467 | |
| Gene (*Drosophila melanogaster*) | Fas3-PG | FlyBase | FLYB: FBpp0311111 | |
| Gene (*Drosophila melanogaster*) | Drice | FlyBase | FLYB: FBpp0084848 | |
| Genetic reagent (*Drosophila melanogaster*) | $w^{1118}$ | **Shinoda et al., 2023** | | Maintained in Masayuki Miura lab |
| Genetic reagent (*Drosophila melanogaster*) | Drice::V5::TurboID | **Shinoda et al., 2019** | FLYB: FBal0356244 | |
| Genetic reagent (*Drosophila melanogaster*) | Dcp-1::V5::TurboID | **Shinoda et al., 2019** | FLYB: FBal0356245 | |
| Genetic reagent (*Drosophila melanogaster*) | Dronc::V5::TurboID | **Shinoda et al., 2019** | FLYB: FBal0356243 | |
| Genetic reagent (*Drosophila melanogaster*) | Pebbled-Gal4 | Bloomington *Drosophila* Stock Center | RRID:BDSC_80570 | |
| Genetic reagent (*Drosophila melanogaster*) | Orb$^{0449}$-Gal4 | Bloomington *Drosophila* Stock Center | RRID:BDSC_63325 | |
| Genetic reagent (*Drosophila melanogaster*) | GH146-Gal4 | Bloomington *Drosophila* Stock Center | RRID:BDSC_30026 | |
| Genetic reagent (*Drosophila melanogaster*) | UAS-Drice-RNAi | Vienna *Drosophila* Resource Center | FLYB: FBst0457273 | |
| Genetic reagent (*Drosophila melanogaster*) | Fas3::EGFP::FlAsH::StrepII::TEV::3xFLAG | Bloomington *Drosophila* Stock Center | RRID:BDSC_59809 | |
| Genetic reagent (*Drosophila melanogaster*) | UASz-Fas3A::3xFLAG | This paper | | Integrated into attP2 landing site |
| Genetic reagent (*Drosophila melanogaster*) | UASz-Fas3B::3xFLAG | This paper | | Integrated into attP2 landing site |
| Genetic reagent (*Drosophila melanogaster*) | UASz-Fas3C::3xFLAG | This paper | | Integrated into attP2 landing site |
| Genetic reagent (*Drosophila melanogaster*) | UASz-Fas3D::3xFLAG | This paper | | Integrated into attP2 landing site |
| Genetic reagent (*Drosophila melanogaster*) | UASz-Fas3G::3xFLAG | This paper | | Integrated into attP2 landing site |

*Appendix 1 Continued on next page*

*Appendix 1 Continued*

| Reagent type (species) or resource | Designation | Source or reference | Identifiers | Additional information |
|---|---|---|---|---|
| Genetic reagent (*Drosophila melanogaster*) | Ubi-FRT-STOP-FRT-mCD8::DQVD::QF2 | This paper | | Integrated into attP40 landing site |
| Genetic reagent (*Drosophila melanogaster*) | Ubi-FRT-STOP-FRT-mCD8::DQVA::QF2 | This paper | | Integrated into attP40 landing site |
| Genetic reagent (*Drosophila melanogaster*) | UAS-FLP | Kyoto Stock Center | DGRC Number: 107788 | |
| Genetic reagent (*Drosophila melanogaster*) | QUAS-tdTomato::3xHA | Bloomington *Drosophila* Stock Center | RRID:BDSC_30005 | |
| Genetic reagent (*Drosophila melanogaster*) | ELAV-Gal4 | Bloomington *Drosophila* Stock Center | RRID:BDSC_8765 | |
| Genetic reagent (*Drosophila melanogaster*) | Or42b-Gal4 | Bloomington *Drosophila* Stock Center | RRID:BDSC_9971 | |
| Genetic reagent (*Drosophila melanogaster*) | UAS-RedStinger | Bloomington *Drosophila* Stock Center | RRID:BDSC_8547 | |
| Genetic reagent (*Drosophila melanogaster*) | UASz-3xFLAG::mNeonGreen | This paper | | Integrated into attP40 landing site |
| Genetic reagent (*Drosophila melanogaster*) | UASz-Drice::myc::mNeonGreen | This paper | | Integrated into attP40 landing site |
| Genetic reagent (*Drosophila melanogaster*) | UASz-V5::TurboID | This paper | | Integrated into attP2 landing site |
| Genetic reagent (*Drosophila melanogaster*) | UASz-Dronc::V5::TurboID | This paper | | Integrated into attP2 landing site |
| Genetic reagent (*Drosophila melanogaster*) | UASz-Dronc$^{CG}$::V5::TurboID | This paper | | Integrated into attP2 landing site |
| Genetic reagent (*Drosophila melanogaster*) | UASz-Dcp-1::V5::TurboID | This paper | | Integrated into attP2 landing site |
| Genetic reagent (*Drosophila melanogaster*) | UASz-Dcp-1$^{CG}$::V5::TurboID | This paper | | Integrated into attP2 landing site |
| Genetic reagent (*Drosophila melanogaster*) | UASz-Drice::V5::TurboID | This paper | | Integrated into attP2 landing site |
| Genetic reagent (*Drosophila melanogaster*) | UASz-Drice$^{CG}$::V5::TurboID | This paper | | Integrated into attP2 landing site |
| Genetic reagent (*Drosophila melanogaster*) | UAS-Diap-1::VENUS | *Koto et al., 2009* | FLYB: FBal0244058 | |
| Genetic reagent (*Drosophila melanogaster*) | UAS-Dronc-RNAi | *Kanuka et al., 2005* | FLYB: FBal0191075 | |
| Genetic reagent (*Drosophila melanogaster*) | UAS-Dronc$^{DN}$::EGFP | *Igaki, 2002* | FLYB: FBal0104748 | |
| Genetic reagent (*Drosophila melanogaster*) | UAS-LacZ-RNAi | *Kennerdell and Carthew, 2000* | FLYB: FBal0196940 | Gift from Dr. Richard Carthew |
| Genetic reagent (*Drosophila melanogaster*) | UAS-Fas3-RNAi#1 | Vienna *Drosophila* Resource Center | FLYB: FBst0471491 | |
| Genetic reagent (*Drosophila melanogaster*) | UAS-Fas3-RNAi#2 | Vienna *Drosophila* Resource Center | FLYB: FBst0458720 | |

*Appendix 1 Continued on next page*

*Appendix 1 Continued*

| Reagent type (species) or resource | Designation | Source or reference | Identifiers | Additional information |
|---|---|---|---|---|
| Genetic reagent (*Drosophila melanogaster*) | UAS-Fas3G-RNAi | This paper | | Integrated into attP40 landing site |
| Cell line (*Drosophila melanogaster*) | S2 | RIKEN BRC | Cat#: RCB1153; RRID:CVCL_Z232 | Maintained in Masayuki Miura lab |
| Antibody | anti-V5 (mouse monoclonal) | Invitrogen | Cat#: 46-0705; RRID:AB_2556564 | WB (1:5,000) |
| Antibody | anti-FLAG M2 (mouse monoclonal) | Sigma | Cat#: F1804; RRID:AB_262044 | WB (1:5,000), IHC (1:500) |
| Antibody | anti-alpha tubulin (DM1A) (mouse monoclonal) | Sigma | Cat#: T9026; RRID:AB_477593 | WB (1:10,000) |
| Antibody | anti-Actin (mouse monoclonal) | Sigma | Cat#: A4700; RRID:AB_476730 | WB (1:5,000) |
| Antibody | anti-Fasciclin 3 (7G10) (mouse monoclonal) | DSHB | Cat#: 7G10 anti-Fasciclin III; RRID:AB_528238 | WB (1:50), IHC (1:20) |
| Antibody | anti-Fasciclin3 isoform G (rabbit polyclonal) | This paper | | WB (1:50) |
| Antibody | anti-Drosophila Lamin B (mouse monoclonal) | DSHB | Cat#: ADL67.10; RRID:AB_528336 | WB (1:1,000) |
| Antibody | anti-GFP (GF090R) (rat monoclonal) | nacalai tesque | Cat#: 04404-26; RRID:AB_10013361 | IHC (1:100) |
| Antibody | anti-mCD8 (rat monoclonal) | Invitrogen | Cat#: MCD0800; RRID:AB_10392843 | IHC (1:50) |
| Antibody | anti-rabbit IgG HRP-conjugated (goat polyclonal) | CST | Cat#: 7074S; RRID:AB_2099233 | WB (1:5,000) |
| Antibody | anti-mouse IgG HRP-conjugated (goat polyclonal) | Promega | Cat#: W402B; RRID:AB_430834 | WB (1:5,000) |
| Antibody | anti-mouse IgG Alexa Fluor 488-conjugated (donkey polyclonal) | ThermoFisher Scientific | Cat#: A-21202; RRID:AB_141607 | IHC (1:500) |
| Antibody | anti-rat IgG Alexa Fluor 488-conjugated (donkey polyclonal) | ThermoFisher Scientific | Cat#: A-21208; RRID:AB_2535794 | IHC (1:500) |
| Antibody | anti-mouse IgG Alexa Fluor 647-conjugated (donkey polyclonal) | ThermoFisher Scientific | Cat#: A-31571; RRID:AB_162542 | IHC (1:500) |
| Antibody | anti-rat IgG Alexa Fluor 633-conjugated (goat polyclonal) | ThermoFisher Scientific | Cat#: A-21094; RRID:AB_2535749 | IHC (1:500) |
| Recombinant DNA reagent | pUASz1.0 (plasmid) | *Drosophila* Genomics Resource Center | Cat#: 1431; RRID:DGRC_1431 | |
| Recombinant DNA reagent | pUASz-Fasciclin 3 isoform A-3xFLAG (plasmid) | This paper | | |
| Recombinant DNA reagent | pUASz-Fasciclin 3 isoform B/F-3xFLAG (plasmid) | This paper | | |
| Recombinant DNA reagent | pUASz-Fasciclin 3 isoform C-3xFLAG (plasmid) | This paper | | |
| Recombinant DNA reagent | pUASz-Fasciclin 3 isoform D/E-3xFLAG (plasmid) | This paper | | |
| Recombinant DNA reagent | pUASz-Fasciclin 3 isoform G-3xFLAG (plasmid) | This paper | | |

*Appendix 1 Continued on next page*

Appendix 1 Continued

| Reagent type (species) or resource | Designation | Source or reference | Identifiers | Additional information |
|---|---|---|---|---|
| Recombinant DNA reagent | pAc5-STABLE2-neo (plasmid) | Addgene | Cat#: 32426; RRID:Addgene_32426 | |
| Recombinant DNA reagent | pAc5-Fasciclin 3 isoform A-3xFLAG (plasmid) | This paper | | |
| Recombinant DNA reagent | pAc5-Fasciclin 3 isoform B/F-3xFLAG (plasmid) | This paper | | |
| Recombinant DNA reagent | pAc5-Fasciclin 3 isoform C-3xFLAG (plasmid) | This paper | | |
| Recombinant DNA reagent | pAc5-Fasciclin 3 isoform D/E-3xFLAG (plasmid) | This paper | | |
| Recombinant DNA reagent | pAc5-Fasciclin 3 isoform G-3xFLAG (plasmid) | This paper | | |
| Recombinant DNA reagent | pAc5-V5-mNeonGreen (plasmid) | *Shinoda et al., 2023* | | |
| Recombinant DNA reagent | pUASz-3xFLAG-mNeonGreen (plasmid) | This paper | | |
| Recombinant DNA reagent | pUASz-Drice-myc-mNeonGreen (plasmid) | This paper | | |
| Recombinant DNA reagent | V5-TurboID-NES_pcDNA3 (plasmid) | Addgene | Cat#: 107169; RRID:Addgene_107169 | |
| Recombinant DNA reagent | pUASz-V5-TurboID (plasmid) | This paper | | |
| Recombinant DNA reagent | pUASz-Dronc-V5-TurboID (plasmid) | This paper | | |
| Recombinant DNA reagent | pUASz-Dronc$^{C318G}$-V5-TurboID (plasmid) | This paper | | |
| Recombinant DNA reagent | pUASz-Dcp-1-V5-TurboID (plasmid) | This paper | | |
| Recombinant DNA reagent | pUASz-Dcp-1$^{C196G}$-V5-TurboID (plasmid) | This paper | | |
| Recombinant DNA reagent | pUASz-Drice-V5-TurboID (plasmid) | This paper | | |
| Recombinant DNA reagent | pUASz-Drice$^{C211G}$-V5-TurboID (plasmid) | This paper | | |
| Recombinant DNA reagent | pJFRC-28K (plasmid) | | | Gift from Dr. Hokuto Kazama |
| Recombinant DNA reagent | pBPQU (plasmid) | *Kashio et al., 2016* | | |
| Recombinant DNA reagent | pJFRC-mCD8-DQVD-QF2 (plasmid) | This paper | | |
| Recombinant DNA reagent | pJFRC-mCD8-DQVA-QF2 (plasmid) | This paper | | |
| Recombinant DNA reagent | pUbi83 (plasmid) | | | Gift from Dr. Daiki Umetsu |
| Recombinant DNA reagent | pJFRC201-10xUAS-FRT-STOP-FRT-myr-smGFP-HA (plasmid) | Addgene | Cat#: 63166; RRID:Addgene_63166 | |
| Recombinant DNA reagent | pUbi-FRT-STOP-FRT-mCD8-DQVD-QF2 (plasmid) | This paper | | |
| Recombinant DNA reagent | pUbi-FRT-STOP-FRT-mCD8-DQVA-QF2 (plasmid) | This paper | | |

Appendix 1 Continued on next page

*Appendix 1 Continued*

| Reagent type (species) or resource | Designation | Source or reference | Identifiers | Additional information |
|---|---|---|---|---|
| Recombinant DNA reagent | pQUAST (plasmid) | Addgene | Cat#: 24349; RRID:Addgene_24349 | |
| Recombinant DNA reagent | pQUAST-mNeonGreen (plasmid) | This paper | | |
| Recombinant DNA reagent | pQUAST-FLPo (plasmid) | Addgene | Cat#: 24357; RRID:Addgene_24357 | |
| Recombinant DNA reagent | pUASz-FLPo (plasmid) | This paper | | |
| Recombinant DNA reagent | pWALIUM20 (plasmid) | *Drosophila* Genomics Resource Center | Cat#: 1472; RRID:DGRC_1472 | |
| Recombinant DNA reagent | pWALIUM20-Fas3G-shRNA (plasmid) | This paper | | |
| Peptide, recombinant protein | The peptide corresponding to the 506–524 aa (N-LKPANEATPATTPAPTTAA-C) of Fasciclin 3 isoform G | This paper | | Generated by Eurofins Inc. |
| Commercial assay or kit | Anti-FLAG M2 Magnetic Beads | Millipore | Cat#: M8823-1ML | |
| Commercial assay or kit | FG-NeutrAvidin beads | Tamagawa Seiki | Cat#: TAS8848 N1171 | |
| Commercial assay or kit | MycoStrip | InvivoGen | Cat#: rep-mys-10 | |
| Commercial assay or kit | Effectene Transfection Reagent | QIAGEN | Cat#: 301427 | |
| Chemical compound, drug | (+)-Biotin | WAKO | Cat#: 029-08713 | |
| Chemical compound, drug | Streptavidin HRP-conjugated | Invitrogen | Cat#: SA10001 | WB (1:10,000) |
| Chemical compound, drug | Hoechst 33342 | Invitrogen | Cat#: H3570 | IHC (8 µM) |
| Chemical compound, drug | Streptavidin-Cy2 | Jackson ImmunoResearch | Cat#: 016-220-084 | IHC (1:500) |
| Chemical compound, drug | Streptavidin-Cy5 | Jackson ImmunoResearch | Cat#: 016-170-084 | IHC (1:500) |
| Chemical compound, drug | SlowFade Gold antifade reagent | Invitrogen | Cat#: S36939 | |
| Chemical compound, drug | Cycloheximide | Sigma | Cat#: C7698 | |
| Chemical compound, drug | Apple cider vinegar | Mizkan | | |
| Software, algorithm | Fiji (ImageJ) | NIH Image | | https://imagej.net/software/fiji/ |
| Software, algorithm | Proteome Discoverer (PD) 2.2 | Thermo Fisher Scientific | | |
| Software, algorithm | DAVID | *Huang et al., 2009* | | https://davidbioinformatics.nih.gov/ |
| Software, algorithm | PSIPRED | | | http://bioinf.cs.ucl.ac.uk/psipred/ |
| Software, algorithm | AlphaFold2-Multimer | *Mirdita et al., 2022* | | ColabFold v1.5.5: AlphaFold2 using MMseqs2 |
| Software, algorithm | GraphPad Prism 8 | GraphPad | | |

