## [Editor Report · eLife Assessment]

This **important** study identifies a mechanism by which caspases are activated in a non-lethal context to induce functional modulation in *Drosophila* olfactory receptor neurons. To deliver, the authors generated a new reporter of caspases, used TurboID to identify proteins proximal of the Drosophila executioner caspases Drice, and then focused on Fasciclin 3 as a mediator. The experimental results and the main conclusions are **convincing**. This substantial body of work will be of interest to researchers across fields, from neuroscience of olfaction to development and cell biology.

---

## [Referee Report · Reviewer #1 (Public review)]

Summary:

In this manuscript, Muramoto and colleagues have examined a mechanism by which the executioner caspase Drice is activated in a non-lethal context in *Drosophila*. The authors have comprehensively examined this in the Drosophila olfactory receptor neurons using sophisticated techniques. In particular, they had to engineer a new reporter by which non-lethal caspase activation could be detected. The authors conducted a proximity labeling experiment and identified Fasciclin 3 as a key protein in this context. While removal of Fascilin 3 did not block non-lethal caspase activation (likely because of redundant mechanisms), its overexpression was sufficient to activate non-lethal caspase activation.

Strengths:

While non-lethal functions of caspases have been reported in several contexts, far less is known about the mechanisms by which caspases are activated in these non-lethal contexts. So, the topic is very timely. The overall detail of this work is impressive and the results, for the most part, are well controlled and justified.

Weaknesses:

The behavioral results shown in Fig. 6 need more explanation and clarification (more details below). As currently shown, the results of Fig. 6 seem uninterpretable. Also, overall presentation of the Figures and description in legends can be improved.

Comments on revisions:

The authors have adequately addressed my comments.

---

## [Referee Report · Reviewer #2 (Public review)]

In this revised version of the study, the authors investigate the role of caspases in neuronal modulation through non-lethal activation. They analyze proximal proteins of executioner caspases using a variety of techniques, including TurboID and a newly developed monitoring system based on Gal4 manipulation, called MASCaT. They demonstrate that overexpression of Fas3G promotes the non-lethal activation of caspase Dronc in olfactory receptor neurons. In addition, they investigate the regulatory mechanisms of non-lethal function of caspase by performing a comprehensive analysis of proximal proteins of executioner caspase Drice. It is important to point out that the authors use an array of techniques from western blot to behavioral experiments and also that the generated several reagents, from fly lines to antibodies. In this revised version of the manuscript the authors addressed the concerns raised by this reviewer in a very thorough way. This is an interesting work that would appeal to readers of multiple disciplines. As a whole these findings suggest that overexpression of Fas3G enhances a non-lethal caspase activation in ORNs, providing a novel experimental model that will allow for exploration of molecular processes that facilitate caspase activation without leading to cell death.

Comments on revisions:

I would like to thank the authors for fully addressing my concerns.

---

## [Author Response]

The following is the authors’ response to the original reviews

**Public Reviews:**

**Reviewer #1 (Public review):**
Summary:In this manuscript, Muramoto and colleagues have examined a mechanism by which the executioner caspase Drice is activated in a non-lethal context in *Drosophila*. The authors have comprehensively examined this in the Drosophila olfactory receptor neurons using sophisticated techniques. In particular, they had to engineer a new reporter by which non-lethal caspase activation could be detected. The authors conducted a proximity labeling experiment and identified Fasciclin 3 as a key protein in this context. While the removal of Fascilin 3 did not block non-lethal caspase activation (likely because of redundant mechanisms), its overexpression was sufficient to activate non-lethal caspase activation.Strengths:While non-lethal functions of caspases have been reported in several contexts, far less is known about the mechanisms by which caspases are activated in these non-lethal contexts. So, the topic is very timely. The overall detail of this work is impressive and the results for the most part are wellcontrolled and justified.Weaknesses:The behavioral results shown in Figure 6 need more explanation and clarification (more details below). As currently shown, the results of Figure 6 seem uninterpretable. Also, overall presentation of the Figures and description in legends can be improved.

We sincerely thank the reviewer for their highly positive evaluation of our study, particularly from a technical perspective. We also greatly appreciate the valuable comments provided on our manuscript. In response, we have revised the manuscript with a particular focus on Figure 6, as well as the overall presentation of the figure and its description in the legends, in accordance with the reviewer’s suggestions. For further clarification, please refer to our detailed point-by-point responses provided below.

**Reviewer #2 (Public review):**
In this study, the authors investigate the role of caspases in neuronal modulation through non-lethal activation. They analyze proximal proteins of executioner caspases using a variety of techniques, including TurboID and a newly developed monitoring system based on Gal4 manipulation, called MASCaT. They demonstrate that overexpression of Fas3G promotes the non-lethal activation of caspase Dronc in olfactory receptor neurons. In addition, they investigate the regulatory mechanisms of non-lethal function of caspase by performing a comprehensive analysis of proximal proteins of executioner caspase Drice. It is important to point out that the authors use an array of techniques from western blot to behavioral experiments and also that the generated several reagents, from fly lines to antibodies.This is an interesting work that would appeal to readers of multiple disciplines. As a whole these findings suggest that overexpression of Fas3G enhances a non-lethal caspase activation in ORNs, providing a novel experimental model that will allow for exploration of molecular processes that facilitate caspase activation without leading to cell death.

We sincerely thank the reviewer for their highly positive evaluation of our study, particularly from a methodological perspective. We also greatly appreciate the valuable comments provided on our manuscript. In response, we have revised the manuscript in line with the reviewer’s suggestions. For further clarification, please refer to our detailed point-by-point responses provided below.

**Reviewing Editor comments:**
I am pleased to let you know that our reviewers found the results in your paper important and the evidence compelling. There are a few minor comments and a point was raised regarding figure 6 for which further details were asked. Please see the reviewer's comments. We are looking forward to receiving an updated version of your very interesting paper.

We are grateful to you and the reviewers for dedicating time to review our manuscript and for providing insightful comments and suggestions. We have revised our manuscript in line with the reviewers' feedback. The major revision involves clarifying the two-choice preference assay presented in Figure 6. Details of these revisions are provided in our point-by-point responses to the reviewers’ comments below. The new and extensively modified sections of text are highlighted in blue. We have introduced new panels (Figures 1D, 3D, 6B, and 6C) and made modifications to Figure 6A. The previous Figure 1D has been relocated to Figure 1–figure supplement 1B. Additionally, our detailed responses to the reviewers’ comments are also highlighted in blue within the point-by-point response section. With all concerns and suggestions from the Editor and reviewers addressed, our conclusion—that executioner caspase is proximal to Fasciclin 3 which facilitates non-lethal activation in *Drosophila* olfactory receptor neurons—is now more robustly supported. We are confident that our revised manuscript makes a significant contribution to the fields of caspase function and neurobiology. We remain hopeful that the reviewers will find it suitable for publication in eLife.

**Reviewer #1 (Recommendations for the authors):**
The main comment here is related to Figure 6, which needs to be better explained. First, if the results in Figure 6B and C are conducted with young flies, why is the preference index close to 0? Aren't these young flies more attracted to ACV? Second, what are the results with Dronc-RNAi and DroncDN alone? These should be shown to more accurately assess the outcome of Fas3G expression with and without Dronc inhibition. Third, if Fas3G overexpression induces non-lethal caspase activation and a behavioral change, why does Dronc inhibition enhance (and not suppress) this behavioral change?

We sincerely thank the reviewer for the comment. We used one-week-old young flies for the two-choice preference assay. We found that 16 hours of starvation combined with 25% ACV in the trap elicited a robust attraction behavior to the vinegar (New Figure 6B). In contrast, 4 hours of starvation with 1% ACV in the trap resulted in milder attraction behavior, with the preference index value being close to 0 but still showing a positive trend (New Figure 6B). Since our hypothesis is that non-lethal caspase activation suppresses attraction behavior, and that inhibiting caspase activation could enhance attraction, we used the milder experimental condition for subsequent analyses.

In the original manuscript, we did not test Dronc inhibition alone because caspase activation is rarely observed in young flies (as demonstrated in Figure 3C, New Figure 3D, etc), suggesting that Dronc inhibition during this stage would not affect behavior. This hypothesis is further supported by previous research showing that inhibition of caspase activity in aged flies restores attraction behavior but does has no effect in young flies (Chihara et al., 2014). To validate this hypothesis, we conducted the two-choice preference assay again, including caspase activity inhibition by *DroncDN* expression alone. As expected, Dronc inhibition alone did not alter behavior in young flies (New Figure 6C).

We also observed that *Fas3G* overexpression promotes a weak, though not statistically significant, enhancement in attraction behavior. Importantly, simultaneous inhibition of caspase activity further enhanced attraction behavior (New Figure 6C). These results suggest that *Fas3G* overexpression has a dual function: one aspect promotes attraction behavior, while the other induces non-lethal caspase activation. In this context, non-lethal caspase activation appears to counteract the behavioral response, acting as a regulatory brake. To address the reviewer’s comments comprehensively, we included the New Figure 6B and replaced the original Figure 6B and C with New Figure 6C. Additionally, we revised the manuscript text as follows:

Using a two-choice preference assay with ACV (Figure 6A), we found that 16 hours of starvation combined with 25% ACV in the trap elicited a robust attraction behavior to the vinegar (Figure 6B). In contrast, 4 hours of starvation with 1% ACV in the trap resulted in milder attraction behavior, with the preference index value being close to 0 but still showing a positive trend (Figure 6B). Under the milder experimental condition, we first confirmed that inhibition of caspase activity through expressing *DroncDN* didn’t affect attraction behavior in young adult (Figure 6C), consistent with a previous report (Chihara et al., 2014).We then observed that the overexpression of *Fas3G*, which activates caspases, did not impair attraction behavior. Instead, it rather appeared to enhance the tendency for attraction behavior (Figure 6C), suggesting that Fas3G promotes attraction behavior. Finally, we found that inhibiting *Fas3G* overexpression-facilitated non-lethal caspase activation by expressing *DroncDN* strongly promoted attraction to ACV (Figure 6C). Overall, these results suggest that *Fas3G* overexpression has a dual function: it enhances attraction behavior while also triggering non-lethal caspase activation, which counteracts the behavioral response, functioning as a regulatory brake without causing cell death.

Other minor comments are below:The authors should clarify that while they refer to their caspases reporters as "non-lethal caspase reporters", these are caspase reporters in general and can report both lethal and non-lethal caspase activation. Of course, the only surviving cells are those that experience non-lethal caspase activation.

We thank the reviewer for pointing this out. This reporter can monitor caspase activation with high sensitivity only if the cell is capable of transcribing and translating the reporter proteins following cleavage of the probe, most likely in living cells. However, as mentioned, using the term “non-lethal reporter” is not accurate, as additional experiments are required to determine whether caspase activation leads to cell death. Therefore, we removed the term “non-lethal” and referred to this reporter simply as a highly sensitive caspase reporter in the revised manuscript.

Some of the figure panels could be better described in the legends (e.g. Figure 1E, 1F, 4E, 4F).

We thank the reviewer for the comment. We have included additional explanations in the figure legends throughout the manuscript.

In Figure 3C, the OL and AL regions should be marked in the figure as done in Figure 1C.

We thank the reviewer for the comment. We have marked OL and AL regions in Figure 3C and Figure 2A as in Figure 1C.

In Figures 4A and B, the authors should rearrange the order of the x-axis to reflect the order that appears in the text (Dronc first).

We thank the reviewer for the comment. We have rearranged the order of labels on the X-axis to reflect the order that appears in the text.

In Figure 6B, do the colors imply anything? If so, it should be explained.

We thank the reviewer for pointing this out. We intended to use the colors where the light blue bars represent *Fas3G* overexpression, while the red dots indicate caspase-activated conditions. In the New Figure 6C, we used light blue dots for *Fas3G* overexpression and red bars for caspase-activated conditions. We have added an explanation in the figure legend. In addition, we have removed the colors in Figure 4B and have added an explanation in the figure legend in Figure 4D.

**Reviewer #2 (Recommendations for the authors):**
(1) For the methods section make a table for the lines, the way they are listed is not the most easy to read.

We thank the reviewer for the comment. We have listed the fly strains used in this study in Table S3.

(2) Lines 420 to 573, not sure why this is here, this information should be in the figure or figure legend, or make a table if necessary.

We thank the reviewer for the comment. We have listed the detailed genotypes corresponding to each figure in Table S4.

(3) Blocking with donkey serum, do you get better results than bovine?

We have not conducted tests with bovine serum for immunohistochemistry. Donkey serum was used throughout the manuscript.

(4) The Methods section is very thorough and complete but I recommend the use of tables to organize some of the reagents used.

We thank the reviewer for the comment. We have listed the fly strains used in this study in Table S3 and the detailed genotypes corresponding to each figure in Table S4.

(5) Line 647 spells out LC-MS/MS.

We thank the reviewer for pointing this out. We have provided the full spelling as “liquidchromatography-tandem mass spectrometry”.

(6) Line 808 spells out ACV (apple cider vinegar) and MQ (MilliQ water).

We thank the reviewer for pointing this out. We have provided the full spelling as suggested.

(7) Figure 1D. Why do you use only females?

We thank the reviewer for pointing this out. In the original manuscript, we analyzed female flies by crossing each Gal4 strain with *UAS-Drice-RNAi; Drice::V5::TurboID* virgin females. In this case, because *Pebbled-Gal4* is located on X chromosome, we could only use female flies for the analysis. To address this, we examined the expression pattern in males flies by crossing each Gal4 virgin female with *UAS-Drice-RNAi; Drice::V5::TurboID* males. As expected, Drice expression is also mostly depleted when using the ORN-specific Gal4 driver, *Pebbled-Gal4*, suggesting that Drice expression is predominantly observed in ORNs in males as well. We have added New Figure 1D to present the male data. The original Figure 1D, which presents female data, has been relocated to Figure 1–figure supplement 1B.

(8) Figure 1D. Be clear about the LN driver used here in the figure.

We thank the reviewer for pointing this out. We used *Orb0449-Gal4* driver (#63325, Bloomington Drosophila Stock Center), which has been previously characterized as an LN-specific Gal4 driver (Wu et al., 2017). Accordingly, we have revised “LN-Gal4” to “*Orb0449-Gal4*” throughout the manuscript.

(9) Figure 1 and Supplementary Figure 1 images are very good. I would recommend the use of a different color palette, to help visualization for colorblind readers (such as this reviewer).

We apologize for any inconvenience caused. We chose the green/magenta color pair because these are complementary colors, which generally provide better contrast compared to other color pairs. Therefore, we have decided to continue using this pair. To enhance readability, we have intensified the magenta signal in the New Figure 1D and Figure 1–figure supplement 1B. We retained the original magenta signal levels in Figure 1C and Figure 1–figure supplement 1A to avoid oversaturation. Instead, we have kept the Streptavidin-only signal images alongside the color merged images for clarity. We hope these adjustments improve the visualization and help you better interpret the figures.

(10) Based on Supplementary Figure 1 and based on the fact that Figures 1B and 1C use males, why not used also males for Figure 1D?

Please refer to our reply to comment #7. We have now included the results for males in the New Figure 1D, which show a similar expression pattern to that observed in females. The results for females originally shown in Figure 1D have been relocated to Figure 1–figure supplement 1B.

(11) Why were the old versus young flies used for Figure 3 raised at 29C? Why not let the animals age at 25C? The use of 29C throughout the manuscript is not clear.

We thank the reviewer for pointing this out. Most of the UAS fly strains used in this study, including a *Fas3G* overexpression line, are *UASz* lines, which exhibit relatively low expression levels compared to UASt lines (DeLuca and Spradling, 2018). Since the Gal4/UAS system is temperature-dependent (Duffy, 2002), we performed most of the experiments at 29°C to enhance gene expression.

For the aging experiments, we chose to rear flies at 29°C because higher temperatures accelerate aging including neuronal aging (Okenve-Ramos et al., 2024), allowing for faster experimentation, and 29°C is within the ecologically relevant range of temperatures for *Drosophila melanogaster* (SotoYéber et al., 2018). Additionally, we confirmed that a subset of olfactory receptor neurons undergo aging-dependent caspase activation at both 29°C and 25°C, as shown in New Figure 3D.

(12) Why not use an Or42b specific GAL 4 for the aging experiment? What are the odorants that are detected by this ORN? Are any of the odorants behaviorally relevant compounds?

We thank the reviewer for pointing this out. While the exact odorant detected by Or42b neurons has not been fully determined, these neurons innervate the DM1 region in the antennal lobe, which is activated by ACV. Additionally, Or42b neurons have been shown to be required for attraction behavior to ACV (Semmelhack and Wang, 2009), supporting the relevance of ACV for the behavioral experiment. We used *Or42b-Gal4* to confirm that Or42b neurons undergo aging-dependent caspase activation, which is detectable using the MASCaT system (New Figure 3D). Furthermore, we verified that these neurons exhibit aging-dependent caspase activation at both 25°C and 29°C (New Figure 3D).

(13) Make the panel lettering in all the figures bigger or bold.

We thank the reviewer for pointing this out. We have increased the size of the panel lettering and made it bold throughout the figures to improve the readability.

(14) Line 806. MilliQ water.

We thank the reviewer for pointing this out. We have ensured that “MilliQ water” is consistently spelled this way throughout the manuscript.

(15) Figure 6. The authors need to be more clear on the experimental conditions. At what time of the day was this experiment performed? Was the experiment run in DD? Were the flies young or old?

We thank the reviewer for pointing this out. We performed the assay using one-week-old young flies under constant dark conditions during both the starvation period and the assay. We have added a detailed explanation in the Methods section. For clarity, we have also revised Figure 6A to provide a more detailed explanation of the experimental setup.

References

Chihara T, Kitabayashi A, Morimoto M, Takeuchi K-I, Masuyama K, Tonoki A, Davis RL, Wang JW, Miura M. 2014. Caspase inhibition in select olfactory neurons restores innate attraction behavior in aged *Drosophila*. PLoS Genet 10:e1004437.

DeLuca SZ, Spradling AC. 2018. Efficient expression of genes in the *Drosophila* germline using a UAS promoter free of interference by Hsp70 piRNAs. Genetics 209:381–387.

Duffy JB. 2002. GAL4 system in *Drosophila*: a fly geneticist’s Swiss army knife. Genesis 34:1–15.

Okenve-Ramos P, Gosling R, Chojnowska-Monga M, Gupta K, Shields S, Alhadyian H, Collie C, Gregory E, Sanchez-Soriano N. 2024. Neuronal ageing is promoted by the decay of the microtubule cytoskeleton. PLoS Biol 22:e3002504.

Semmelhack JL, Wang JW. 2009. Select *Drosophila* glomeruli mediate innate olfactory attraction and aversion. Nature 459:218–223.

Soto-Yéber L, Soto-Ortiz J, Godoy P, Godoy-Herrera R. 2018. The behavior of adult *Drosophila* in the wild. PLoS One 13:e0209917.

Wu B, Li J, Chou Y-H, Luginbuhl D, Luo L. 2017. Fibroblast growth factor signaling instructs ensheathing glia wrapping of *Drosophila* olfactory glomeruli. Proc Natl Acad Sci U S A 114:7505–7512.